# The third Met Office Unified Model-JULES Regional Atmosphere and Land Configuration, RAL3

Mike Bush[1], David L.A. Flack[1], Huw W. Lewis[1], Sylvia I. Bohnenstengel[2], Chris J. Short[1], Charmaine Franklin[3], Adrian P. Lock[1], Martin Best[1], Paul Field[1], Anne McCabe[1], Kwinten Van Weverberg[1,7], Segolene Berthou[1], Ian Boutle[1], Jennifer K. Brooke[1], Seb Cole[1], Shaun Cooper[3], Gareth Dow[1], John Edwards[1], Anke Finnenkoetter[1], Kalli Furtado[4], Kate Halladay[1], Kirsty Hanley[2], Margaret A. Hendry[1], Adrian Hill[1], Aravindakshan Jayakumar[5], Richard W. Jones[1], Humphrey Lean[2], Joshua C.K. Lee[4], Andy Malcolm[1], Marion Mittermaier[1], Saji Mohandas[5], Stuart Moore[6], Cyril Morcrette[1], Rachel North[1], Aurore Porson[2], Susan Rennie[3], Nigel Roberts[2], Belinda Roux[3], Claudio Sanchez[1], Chun-Hsu Su[3], Simon Tucker[1], Simon Vosper[1], David Walters[1], James Warner[1], Stuart Webster[1], Mark Weeks[1], Jonathan Wilkinson[1], Michael Whitall[1], Keith D. Williams[1], and Hugh Zhang[4]

[1]Met Office, FitzRoy Road, Exeter, EX1 3PB, UK
[2]MetOffice@Reading, Brian Hoskins building, Earley Gate, University of Reading, Reading, RG6 6BB, UK
[3]Bureau of Meteorology, Melbourne, Victoria, Australia
[4]Meteorological Service Singapore (MSS), PO Box 8, Changi Airport, Singapore 918141
[5]National Centre for Medium Range Weather Forecasting (NCMRWF), Noida, India
[6]National Institute of Water & Atmospheric Research Ltd (NIWA), 301 Evans Bay Parade, Greta Point, Wellington, 6021, New Zealand
[7]now at Royal Meteorological Institute of Belgium, Brussels, Belgium and Department of Geography, Ghent University, Ghent, Belgium

**Correspondence:** Mike Bush (mike.bush@metoffice.gov.uk)

**Abstract.**

The third version of the Regional Atmosphere and Land (RAL3) science configuration is documented. Developed through international partnership, RAL configurations define settings for the Unified Model atmosphere and Joint UK Land Environment Simulator when applied across timescales with kilometre and sub-km scale model grids. The RAL3 configuration represents a major advance compared to previous versions by delivering a common science definition suitable for application to tropical and mid-latitude regions. Developments within RAL3 include the introduction of a double-moment microphysics scheme and a bi-modal cloud scheme, replacing use of a single-moment scheme and different cloud schemes for mid-latitudes and tropics in previous versions. Updates have been implemented to the boundary layer scheme and a consolidation of land model settings to be more consistent with Global Atmosphere and Land (GAL) science configurations. Physics developments aimed to address priorities for model performance improvement identified by users. This paper documents the RAL3 science configuration, including a series of iterative revisions delivered since it's first release, and their characteristics. Evidence is provided from the variety of assessments of RAL3, relative to the previous version (RAL2). Collaborative development and evaluation across organisations has enabled evaluation across a range of domains, grid-spacing and timescales. The analysis indicates more realistic precipitation distributions; improved representation of clouds and of visibility; a continued trend to more

realistic representation of convection; and reduced near-surface wind speeds, but a persistent cold temperature bias. Overall the convective-scale verification scores and climatological model distributions relative to observations improve for the majority of variables. Ensemble results show improvements to the spread-error relationship. User feedback from subjective assessment activities has also been positive. Differences between RAL3 revisions and RAL2 are further illustrated through process-based analysis of a convective system over the UK. The latest RAL3 configuration (RAL3.3) is therefore recommended for research, operational numerical weather prediction and climate production at km and sub-km scales.

## 1 Introduction

A coordinated process has been implemented over recent years for the development, evaluation, definition and release of standard science configurations for Regional Atmosphere and Land (RAL) model applications of the Met Office Unified Model atmosphere (UM; Brown et al., 2012) and Joint UK Land Environment Simulator (JULES; Best et al., 2011; Clark et al., 2011). Starting with the first (RAL1; Bush et al., 2020) and second (RAL2; Bush et al., 2023) releases, common defined science configurations are routinely and consistently applied across UM/JULES research, operational NWP and climate applications at kilometre-scale. These releases have been delivered through collaboration across the Momentum Partnership (previously UM Partnership) of operational and research centres who use and develop UM/JULES-based prediction systems. The Momentum Partnership includes the Met Office in the UK, the Bureau of Meteorology in Australia, Meteorological Service Singapore (MSS), the National Centre for Medium Range Weather Forecasting (NCMRWF) in India, and the National Institute of Water and Atmospheric Research (NIWA) in New Zealand. This paper documents the third RAL science configuration, termed RAL3, for kilometre and sub-kilometre-scale modelling using the UM and JULES.

While not fully resolving atmospheric deep convection, numerical models with horizontal grid-lengths of the order of a kilometre are able to explicitly represent many of the key dynamical convective processes. Several aspects of mesoscale phenomena, such as convection are better represented when explicitly modelled than when parametrized (as required in coarser-resolution models), even though the correct scales of individual convective cells will not be resolved in general (e.g. Clark et al., 2016). For example, the diurnal cycle of convection over land, the life-cycle and advection of convective clouds and organisation of convective systems are all generally better represented. Such models with kilometre-scale grid lengths are commonly referred to as convection permitting (CP).

It has become standard practice for national meteorological and hydrological services and research organisations to make use of CP regional atmospheric models within their prediction systems (e.g. Baldauf et al., 2011; Tang et al., 2013; Brousseau et al., 2016; Bengtsson et al., 2017; Klasa et al., 2018). These provide valuable information on local weather and in particular high-impact weather, whose prediction is of critical importance to society. In addition to numerical weather prediction (NWP), CP models are an important tool for understanding how weather hazards and resources will change in future (e.g. Prein et al., 2015). While multi-decadal regional CP modelling requires significant computational resources, by explicitly representing convective systems these simulations can provide actionable information to society such as on likely changes to the frequency of intense precipitation (e.g. Kendon et al., 2017). The key motivations for maintaining and developing a capability for regional CP modelling are expanded below.

*Numerical Weather Prediction*: Orographic and coastal effects on weather are more accurately represented at the km and sub-km resolutions of CP models, allowing for greater spatial and temporal detail in forecasts and more accurate prediction of related extremes such as in orographically enhanced rainfall, or temperature minima in valley cold pools. Small-scale processes such as gravity waves and convection can be explicitly represented, avoiding the dependency on physical parametrization, and CP models tend to have more realistic development and propagation of mesoscale convective systems (e.g. Schumacher and Rasmussen, 2020). Improved convective characteristics are important in the tropics where precipitation is dominated by convection and convection-parametrizing model precipitation tends to be unrealistically smoothed, with more widespread low precipitation rates than observed (e.g. Vogel et al., 2020). Simulation on the scale of river catchments afforded by CP models allows the prospect for flood warnings, and CP model output can be directly coupled to local hazard impact models. CP models are therefore a valuable tool for Operational Meteorologists in issuing warnings and guidance.

*Climate Downscaling*: While coarser resolution global climate models will continue to be the major tool for future projections of the Earth System, provided sufficient computing resources are available, kilometre-scale regional climate predictions can be run over multi-annual to multi-decadal timescales. These simulations support a number of goals (e.g. Prein et al., 2015), including to deliver national-scale climate projections; deliver new guidance and driving data for regional impacts modelling; support policy making decisions, particularly related to climate risk and adaptation; assess the extent to which currently available regional climate projections from coarser resolution models are robust for given variables or applications. While CP climate models do not necessarily better represent daily mean precipitation (e.g. Berthou et al., 2020), they typically have improved sub-daily rainfall characteristics including representation of the diurnal cycle of convection, spatial structure of rainfall, duration-intensity characteristics and intensity of hourly precipitation extremes than in climate models with parametrized convection (Kendon et al., 2017). There is clear added value from CP models for simulating current and future climate where deep convection is dominant and in regions of spatial heterogeneity such as mountains and urban areas (e.g. Ban et al., 2021), and in providing tools to examine storm characteristics in a future climate (e.g. Manning et al., 2022).

*Process Research and Model Development*: Alongside cloud-resolving large-eddy simulation (LES), CP models are a powerful tool for understanding atmospheric processes. Relative to typical resolution global models, CP simulations offer a "laboratory" with which to simulate the behaviour of the atmosphere in complex situations and over relative large regions (typically larger than LES) with unprecedented detail leading to new insights. For example, CP models have been used to understand

the links between trade-wind cumulus clouds, their organization, and larger scales - a large source of uncertainty in climate projections (Saffin et al., 2023); the influence of the land surface on convection (Henderson et al., 2022); the impacts of orography on large-scale momentum budgets (Sandu et al., 2019). This analysis requires confidence in a model to realistically represent the small-scale processes of interest, and inclusion of sufficient model process complexity. Assuring confidence in model performance is in itself a challenge, requiring high-resolution observations and development of diagnostic methods to evaluate the models. Understanding from CP models can also often inform inform physical parametrization development in coarser resolution global and regional models, in which convection is parametrized (e.g. Lavender et al., 2024). Use of CP configurations in models with sub-km grid spacing, sometimes termed "hectometric" models (Lean et al., 2024), can provide insight on the resolution-dependence of convective processes. In addition, sub-km models are becoming increasingly viable tools with which to provide information on urban-areas, either as operational prediction systems (e.g. Theethai Jacob et al., 2023) or to underpin machine learning for local information (e.g. Blunn et al., 2024).

*Developing Future Capability*: Understanding the characteristics and limitations of current CP models and developing improved model configurations is essential to improve their utility. Their improvement is particularly important given growing interest in application of CP models globally, enabled by exploiting exascale computing platforms (e.g. Slingo et al., 2022; Hohenegger et al., 2022; Stevens et al., 2019). While the focus of environmental reanalysis of observations to date has tended to focus on provision of global-scale information, a number of applications for specific regions have tended to be built on limited-area modelling and data assimilation foundations (e.g. Ridal et al., 2024; Su et al., 2021; Rani et al., 2021). For example, Rasmussen et al. (2023) highlight the opportunities presented from CP reanalyses to generate high-resolution, self-consistent, long-term datasets appropriate for forcing catchment-scale hydrological models. The rapid evolution of reanalysis and simulation trained machine learning approaches for weather and climate prediction significantly amplifies this potential, and the importance of robust underpinning CP models. Regional CP models also provide the basis for development of multi-component environmental prediction frameworks in which atmosphere models are coupled with land, hydrological, ocean and wave models, and potentially also atmospheric chemistry and ocean biogeochemistry components (e.g. Lewis et al., 2019; Warner et al., 2010). CP coupled modelling systems may offer improvements in both local-scale predictive skill, through representation of feedbacks between environmental systems, and also provide new capability to deliver more consistent information to users on compound environmental impacts and natural hazards.

Given the range of applications of CP models, defining a single configuration that performs effectively in all regions has been a long term goal for the Momentum Partnership. For RAL1 and RAL2, depending on the region of interest, an appropriate choice of either mid-latitude RAL-M or tropical RAL-T definitions had to be made, so for example the Bureau of Meteorology use RAL1-T for operational forecasting for a domain centred on Darwin in the tropics and RAL1-M for a domain centred on Melbourne with a more mid-latitude climate. The Met Office currently uses RAL2-M for operational NWP over the UK using a deterministic forecast system (UKV; Tang et al., 2013), with horizontal grid-lengths of 1.5 km and ensemble prediction systems (MOGREPS-UK; Porson et al., 2020) with grids of 2.2 km. For regional climate projection, kilometre-scale simulations have been run with horizontal grid-lengths of 1.5 km over the United Kingdom (Kendon et al., 2014, 2017), 2.2 km over Europe (Berthou et al., 2020) and 4.4 km over Africa (Stratton et al., 2018). The RAL configuration is also applied for sub-km

implementations of the UM, for example focussed on urban applications (e.g Hanley and Lean, 2024; Theethai Jacob et al., 2023; Boutle et al., 2016). With RAL3 this long-term goal has been achieved.

The RAL3 science configuration is defined in Section 2, with revisions described in Section 3. Evaluation of the performance of RAL3 relative to observations and the previous model configuration is discussed in Section 4, with analysis illustrated across different timescales and a number of regions around the world with contrasting meteorology. Section 5 provides a more process-oriented analysis of differences between RAL3 and RAL2 through considering a case study of a quasi-linear convective system over the UK. Computational aspects are presented in Section 6 and conclusions and future development priorities discussed in Section 7.

## 2 Defining Regional Atmosphere and Land - version 3 (RAL3)

### 2.1 Configuration development and assessment process

Building on previous RAL configurations, RAL3 delivers a unified model definition for mid-latitude and tropical application. Developments within RAL3 include the introduction of a double-moment microphyics scheme (Field et al., 2023), a bimodal cloud scheme (Van Weverberg et al., 2021a, b), updates to the boundary layer and land surface options and parameters, and review and removal of differences between mid-latitude and tropical definitions previously required for RAL2 (Bush et al., 2023).

A number of "packages" of candidate options were considered and assessed prior to definition of the released RAL3 configuration, making use of an online repository-based ticket tracking system to support documentation of individual changes to the model and their separate impacts on model performance in initial research testing. Separate tickets are assigned to each model development to provide clarity on which changes are included within a given configuration. Evidence on model performance across the range of tests conducted was provided to a RAL Governance Group consisting of developers and users for review and decision-making prior to definition and release of a configuration, and any subsequent revisions. Table 1 provides an overview of RAL3 developments relative to the previous RAL2-M and RAL2-T definitions.

In addition to continued assessment of shorter-duration case studies and NWP trials (as in previous RAL development cycles), the RAL3 evaluation process introduced multi-annual "climate" testing for the first time. Results from short climate simulations have formed a critical part of the evidence base to support decision-making in the development of RAL3 and subsequent characterization of its performance.

Ensemble predictions are also essential to convective-scale modelling, providing a measure of the uncertainty in the forecast evolution and local-scale details to the initial conditions, lateral boundaries and model physics. The impact of RAL science upgrades on the ensemble performance had previously only been considered towards the end of each development cycle, leaving only a short amount of time to resolve specific issues before making the science upgrade operational. For RAL3 development, ensemble trials over two representative domains (UK and Darwin, Australia) were run earlier in the process to support evaluation of configuration options.

Testing of proposed configurations across multiple domains in different parts of the world including mid-latitude and tropical regions continues to be a defining aspect of RAL development. Extensive testing is only achievable by ever closer collaboration across the Momentum Partnership. Collaboration includes applying common agreed verification metrics across multiple testing and evaluation activities to ensure comparability of results across domains and configurations (see Section 4).

### 2.2 RAL3 release and revisions

Following initial release of the defined RAL3 configuration in summer 2022, further application and its ongoing assessment highlighted a need to address identified issues (Section 3). Some issues became apparent through testing across a wider range of weather conditions. The need to improve some issues through releasing revisions to the initially defined RAL3 configuration following relatively short development cycles was motivated by ongoing assessment towards operational implementations and addressing the needs of users. Potential solutions were tested and evaluated for a sub-set of simulations, critically including multiple domains, with at least case study and multi-annual (climate) simulations required prior to agreement on a new defined revision. As described in Section 3, the initial RAL3 release was re-labelled RAL3.0 for clarity, and subsequent updates to the RAL3 definition have been delivered through revisions, identified as RAL3.1, RAL3.2 and RAL3.3. Decision-making continues to be overseen by the RAL Governance Group.

### 2.3 Dynamical Core: Horizontal and vertical grid

The primary atmospheric prognostic variables are the three-dimensional wind components, virtual dry potential temperature ($\Theta_{vd}$), Exner pressure, dry density, six moist prognostics, and aerosol. Variables are discretized horizontally onto a longitude-latitude grid with Arakawa C-grid staggering (Arakawa and Lamb, 1977). In the vertical the prognostic variables are computed on levels with Charney-Phillips staggering (Charney and Phillips, 1953). A terrain-following hybrid height coordinate is used that blends between flat altitude-based levels and terrain following levels towards the surface (Davies et al., 2005). RAL3 is defined with a 90-level vertical level set which has 67 levels below $18\,\mathrm{km}$ and a fixed model lid $40\,\mathrm{km}$ above sea level. The lowest vertical model level is set at 2.5 m above the surface for wind variables, and 5 m for scalar variables.

### 2.4 Dynamical Core: spatio-temporal discretization

RAL3 uses the UM ENDGame dynamical core, which uses a semi-implicit (SI) time-stepping and semi-Lagrangian (SL) advection formulation to solve the non-hydrostatic, fully-compressible deep-atmosphere equations of motion (Wood et al., 2014). The discrete equations are solved using a nested iterative structure for each atmospheric time step within which some terms are lagged and computed in an outer loop, while others are treated quasi-fully implicitly in an inner loop. The SL departure point equations are solved within the outer loop using a centred average of the previous time step (time $n$) wind and the latest estimates for the current time step (time $(n+1)$) wind. Each of the prognostic variables are interpolated to

**Table 1.** Summary of developments introduced to RAL3 relative to RAL2-M and RAL2-T. Items listed with RAL3 revision numbers indicate where a development was introduced subsequent to the first RAL3 release.

| Process | RAL ticket | Summary description of RAL3 relative to RAL2 | Sub-section |
|---|---|---|---|
| Advection | #62 | Posteriori Monotone Filter (PMF) used for both potential temperature and moisture | 2.4 |
| | #95 | Low-order cubic Hermite (LOCH) scheme used for both potential temperature and moisture | 2.4 |
| | #81 | Fountain buster scheme introduced to correct locally convergent flow | 2.4 |
| | #460 | Change setting for moisture and potential temperature from PMF to tri-linear Lagrange | 3.3 *(RAL3.3)* |
| Microphysics | #192 | 2-moment CASIM (Field et al., 2023), replacing 1-moment Wilson and Ballard (1999) | 2.6 |
| | #189 | Corrections to thunderstorm electrification scheme (McCaul et al., 2009) | 2.6 |
| | #375 | Introduction of CFL-like limit to prevent excessive cloud number removal ('radar holes') | 3.1 *(RAL3.1)* |
| | #471 | Enable washout of Murk aerosol in precipitation | 3.3 (RAL3.3) |
| | #478 | Introduce maximum size of snow prior to changing particle size distribution | 3.3 *(RAL3.3)* |
| | #479 | Improve spin-up for hydrometeors in lateral boundary conditions | 3.3 *(RAL3.3)* |
| Cloud | #83,#111 | Bi-modal cloud scheme (Van Weverberg et al., 2021b), replacing Smith (1990) in RAL2-M and PC2 (Wilson et al., 2008) in RAL2-T | 2.7 |
| BL | #200 | Revised blending between 3D Smagorinsky and 1D BL schemes above diagnosed BL | 2.8 |
| | #530 | Remove time-correlated stochastic BL perturbations to moisture and temperature | 2.8 |
| | #70 | Revision of shear-dominated boundary layer diagnosis | 2.8 |
| | #63 | Diagnose mixing length and convergence of RAL2-M and RAL2-T diffusion settings | 2.8 |
| | #87 | Revision of turbulent kinetic energy and variance diagnostics | 2.8 |
| | #283 | Enable representation of 'frictional' heating arising from dissipation of turbulence | 2.8 |
| Radiation | #473 | Update radiation scheme to be more consistent with CASIM | 3.3 *(RAL3.3)* |
| Orography | #59 | Include turbulent form drag from flow over complex terrain, following GAL approach | 2.8 |
| Land surface | #84 | Review of regional land settings (Bush et al., 2023) to be more consistent with GAL schemes and parameters (see Table 2) | 2.9 |
| | #89 | Updated ancillary information and parameter settings for surface albedo and evapotranspiration | 2.9 |
| | #74 | Correction to couple urban roof tile radiatively to the soil in MORUSES urban scheme | 2.9 |
| | #98 | Correction to enable graupel to pass through vegetation canopy to surface | 2.9 |
| | #311 | Restructure surface exchange code to avoid 'hot spots' | 3.2 *(RAL3.2)* |
| Diagnostics | #97 | Add option to use aerosol climatologies within visibility calculation (when Murk not in use) | 2.6 |
| | #604 | Optimising compuational efficiency of calling visibility diagnostic with CASIM | 3.3 *(RAL3.3)* |
| | #605 | Correction to wind gust diagnostic | 3.3 *(RAL3.3)* |

its appropriate departure point using Lagrange interpolation, with a variety of options of polynomial order. Since pointwise

Lagrangian interpolation is not conservative, the Zero Lateral Flux (ZLF) scheme of Zerroukat and Shipway (2017) was introduced at RAL1 (Bush et al., 2020) to enforce conservation of the mass of dry air, various water species, and any other transported tracers, within the model domain.

RAL3 introduces three key changes to address the non-conservative behaviour of SL advection.

i Extend the use of the Posteriori Monotone Filter (PMF) monotonicity scheme to potential temperature (RAL Ticket #62). This scheme was implemented for moisture variables only in RAL1 and RAL2 as part of the ZLF scheme (Zerroukat and Shipway, 2017). This change was found to substantially alleviate a problem of recurrent instances of dry grid-point storms in the operational UK NWP model.

ii Use the Low-Order Cubic Hermite (LOCH) scheme, a semi-Lagrangian interpolation option, for both $\Theta_{vd}$ and moisture (RAL Ticket #95). Previously RAL2 only used this for $\Theta_{vd}$. This change reduces the error of the semi-Lagrangian scheme in regions where there is a simultaneous occurrence of a lack of smoothness in the advected field and the presence of wave motion, causing the vertical wind to reverse sign periodically.

iii Implement a "fountain buster" scheme (RAL Ticket #81). This addition modifies the UM SL advection scheme and is aimed at suppressing "eternal fountains" whereby single grid-column updrafts can become unrealistically intense and persistent because a stagnation point forms at the base of the updraft. The scheme works by identifying grid-points where the winds are converging into a stagnation point and applies a simple linear up-wind advection increment to add in the effects of convergent in-flow missed by the SL advection. This change has been shown to improve locally convergent flow into updraughts that is underestimated by SL advection, thereby giving more realistic turbulent structures (Lock et al., 2024).

## 2.5 Shortwave and Longwave Radiation

No changes to the radiation parametrization or parameters had been implemented between RAL2 and the initial RAL3 configuration (but see also Section 3.3 for later revision). The SOCRATES radiative transfer scheme (Edwards and Slingo, 1996; Manners et al., 2018) is used with a configuration based on GA3.1 (Walters et al., 2011) to compute Shortwave (SW) and Longwave (LW) radiation components, and provide atmosphere temperature increments that are applied to the prognosed temperature and surface fluxes, and used to derive model diagnostic fluxes. Solar radiation is treated in six SW spectral bands and thermal radiation in nine LW bands. An approximate treatment of scattering is used in computing LW (Manners et al., 2018) to reduce run time. The treatment of gaseous absorption was significantly updated at RAL1 (Bush et al., 2020) to be consistent with the configuration used with GA7 (Walters et al., 2019).

### 2.6 Microphysics

Cloud microphysics parametrizations control the transition of water between phases and hydrometeor species. The Cloud AeroSol Interacting Microphysics scheme (CASIM; Field et al., 2023) is introduced in RAL3 (RAL Ticket #192). CASIM is an open-source, configurable, multi-moment bulk microphysics scheme which can represent cloud microphysics processes and aerosol-cloud interactions across spatial and temporal scales and has therefore been designed to interface with atmospheric models of differing dynamic complexity. Users can define the number of cloud species and the associated number of moments. In RAL3, CASIM represents cloud by using 5 species (cloud liquid, rain, ice, snow, and graupel). Since CASIM is a multi-moment scheme, these species can all be specified by one prognostic moment (mass mixing ratio) or two moments (mass mixing ratios and species number concentration). In addition, rain, snow, and graupel can be represented with 3 prognostic moments (mass mixing ratios, number concentration, and size distribution "shape"). In RAL3 CASIM is implemented as a two-moment scheme, and it uses a fixed in-cloud number concentration. Because a fixed in-cloud number concentration is being used it is necessary to taper the concentration when going higher in the troposphere to avoid the production of large numbers of small ice crystals via ice nucleation that will result in persistent thick cirrus cloud. An exponential decay is applied that reduces the in-cloud droplet number concentration above 2 km with an e-folding length scale of 2500 m. There is also a downward linear taper to capture the reduced droplet number concentrations near the surface for fog. This taper starts at 50 m and linearly reduced the droplet concentration to 20 per kg in the lowest layer. Aerosol climatologies or the Murk scheme (Clark et al., 2008) are used for diagnosing visibility (RAL Ticket #97).

In addition to a new microphysics scheme, changes to the thunderstorm electrification (lighting) scheme of McCaul et al. (2009) have been introduced (RAL Ticket #189). These include review of the McCaul et al. (2009) implementation in the UM to use correct units, correcting a graupel autoconversion bug that caused spurious graupel production from sublimating ice, and modification of definitions for lightning-producing storms based on whether a graupel water path or total ice water path threshold are exceeded. A review of the tuneable parameters was initially tested over India and following an assessment over the UK, new values for coefficients $k_1$=0.21 and $k_2$=0.60 are recommended for all regions of the world, rather than either regionally-tuned or use of the RAL2 default values of $k_1$=0.042 and $k_2$=0.20.

### 2.7 Cloud

Prior to RAL3, the RAL-T configurations performed well for deep convective environments and used the PC2 prognostic cloud fraction scheme (Wilson et al., 2008). However, this configuration tends to under-predict low cloud cover in overcast situations, so less well suited to CP modelling in mid-latitudes. The RAL-M configurations therefore used the diagnostic Smith cloud fraction scheme (Smith, 1990) which represents low cloud over the mid-latitudes much better than RAL-T, but struggled with organized deep convection typical of the tropics. The good low cloud performance of the RAL-M can to a large extent be attributed to empirical bias corrections that are easier to apply in a diagnostic framework such as the Smith scheme.

Inherently, the Smith scheme also struggles with low cloud, since it diagnoses cloud fraction on the assumption that humidity deviations within a model grid box are symmetric around the mean, and uni-modal. Evidence from aircraft and ground-based lidar observations demonstrate that the humidity variability in the atmosphere is seldom symmetric and uni-modal (Wood and Field, 2000; Turner et al., 2014; Wulfmeyer et al., 2016). For example, stratocumulus typically resides just below temperature inversions, where humidity variations tend to be negatively skewed and even bimodal. Distributions consist of a broad mode of moist boundary-layer air, and a less pronounced dry mode originating from occasional intrusions of free-tropospheric air detrained below the general inversion level. These two air masses do not mix readily and can co-exist within a volume typical of a model grid box.

A new cloud scheme has been developed at the Met Office and introduced in RAL3 that can account for this bimodal behaviour near the boundary-layer top (Van Weverberg et al., 2021b, a) (RAL Ticket #83, #111). It first identifies entrainment zones associated with sharp temperature inversions. Within these entrainment zones a bimodal distribution of humidity deviations is reconstructed. The distribution includes a mode of dry air from the top of the entrainment zone, and a second mode of moist air from the bottom of the entrainment zone and uses their respective turbulent properties in calculating cloud liquid water content and cloud fraction, conserving the grid-box mean value of saturation departure.

Based on evaluation of its performance for a six-week intensive observation campaign near the Atmospheric Radiation Measurement (ARM) facility on the Southern Great Plains in USA, Van Weverberg et al. (2021a) found that the bimodal cloud scheme improved cloud cover compared to diagnostic and prognostic cloud schemes that rely on a uni-modal, symmetric sub-grid humidity distribution and better represented the relationships between cloud cover, liquid water content and relative humidity. It outperformed current configurations for cloud water content and radiative and optical properties, in particular for stratocumulus. Ice cloud fraction representation in the bimodal parametrization is the same as in RAL2-M scheme, as described by (Abel et al., 2017).

### 2.8 Boundary Layer (BL)

The parametrization of turbulent motions in kilometre-scale models requires careful treatment because, although most turbulent motions are unresolved, the largest scales can be of a similar size to the grid-length. CP models must therefore be able to parametrize the smaller scales, resolve the largest, and not alias turbulent motions smaller than the grid-scale onto the grid-scale. The "blended" boundary-layer (BL) parametrization described by Boutle et al. (2014) is used to achieve this. In RAL3, this transitions from a 1D vertical turbulent mixing scheme (Lock et al., 2000), suitable for coarser resolution simulations, more typically used in global models, to a 3D turbulent mixing scheme more suited to convective-scale and turbulence permitting simulations based on Smagorinsky (1963). The relative weight given to the 1D versus 3D scheme depends on the ratio of the grid-length to a diagnosed turbulent length scale. The resulting blended eddy diffusivity is applied to down-gradient mixing in all dimensions, whilst appropriately weighted non-local fluxes of heat and momentum are retained in the vertical for unstable boundary-layers. A number of changes introduced to this scheme at RAL3 are described below.

i To harmonise the turbulent mixing settings in RAL3 (RAL Ticket #200), the blended mixing length used in non-turbulent regions of the free troposphere was greatly reduced relative to RAL2. The original philosophy of Boutle et al. (2014) was, in the absence of other information, to blend with height above the boundary layer towards the Smagorinsky mixing length (that is related to the grid size and so is relatively large for typical CP simulations). Initial tests with the new bimodal cloud scheme, which uses the turbulent mixing strength to diagnose subgrid moisture variability, found the use of this large mixing length could lead to spurious diagnosis of liquid cloud in the upper troposphere. For RAL3 the approach is to assume the turbulence length scale will be small in quiescent air and so blend to the 1D mixing length (which has a background value of 40 m) in non-turbulent layers above the boundary layer.

ii Turbulence settings in RAL2-M and RAL2-T were reviewed, and a combined approach defined (RAL Ticket #63). RAL3 therefore uses the conventional unstable stability functions (as used in RAL2-M), but diagnoses mixing lengths in elevated turbulent layers (with subcritical Richardson number, as used in RAL2-T). In RAL3, turbulent layer depths are diagnosed through the atmospheric column and a mixing length $L$ defined as whichever is larger of $0.15$ times the turbulent layer depth or 40 m. Stochastic perturbations in the boundary layer (used only in RAL2-M) were turned off and the performance of initial package testing found to be acceptable for initiation of convection in both mid-latitude and tropical domains. Furthermore, not having the perturbations in mid-latitudes was found to improve convective organisation later in the day.

iii A revision to the diagnosis of BL type has been introduced (RAL Ticket #70), which aligns RAL3 with the Global Atmosphere approach from GA8. In the Lock et al. (2000) scheme, an estimate of the BL depth is diagnosed as the height where a moist adiabatic parcel becomes negatively buoyant. That layer can then be diagnosed as cumulus capped, based on vertical moisture gradients. In regimes of significant wind shear, a shear-dominated BL type is diagnosed based on a threshold ratio of the boundary layer height to the Obukhov length. For those model columns, the non-local scheme and the diagnosis of cumulus (if applicable) are turned off and a local Richardson number BL scheme is used instead. For RAL3 a restriction is introduced to only allow this shear-dominated BL type to be diagnosed in regimes identified as cumulus capped, with the non-local scheme continued to be used in BL regimes identified as well-mixed. This change was targeted mainly at improving predictions in cold air outbreaks but affects other regimes too.

iv The heating arising from the dissipation of turbulence has been included (RAL Ticket #283). Although an observed physical effect, and parametrized in Global UM configurations, this process has previously been omitted from uncoupled RAL configurations as UM-based CP models tended to over-deepen tropical cyclones when this process was included. Castillo et al. (2022) demonstrated the importance of representing this process in regional coupled simulations using the UM, and in preliminary tests of including it in RAL3 it was found that that the small resulting deepening of tropical cyclones was beneficial and that there was little impact on other metrics of performance. This change further aligns RAL3 with global configuration settings.

Geoscientific Model Development Discussions Open Access EGU

v Turbulent kinetic energy (TKE) and variance diagnostics have been revised (RAL Ticket #87). Diagnostics for TKE and variances of vertical velocity, temperature and humidity in RAL3 now directly use scalar fluxes rather than being represented as a down-gradient diffusion. This avoids a side-effect of the non-local BL scheme parametrizing entrainment fluxes across sharp inversions which leads to the diffusion coefficients being set to zero.

vi Turbulent form drag arising from flow over complex terrain is now parametrized in RAL3 (RAL Ticket #59). This had been included in global UM configurations for some time but had not been implemented in RAL2. Previous UK regional NWP trials using a distributed drag parametrization showed a degradation of the near-surface wind speed and temperature errors, where the slowing of the winds reduced boundary layer turbulence leading to excessive surface cooling at night. Because this change interacts strongly with the drag from surface vegetation they were combined for GAL8 (as described in Williams et al. (2020)) and for RAL3 implementation initially tested alongside broader land surface changes (RAL Ticket #84; see section 2.9).

### 2.9 Land Surface

Exchanges of mass, momentum and energy between the atmosphere and the underlying land and sea surfaces are represented using the community land surface model JULES (Best et al. (2011); Clark et al. (2011)). In keeping with the seamless approach to model development, the configuration adopted in RAL3 largely follows that of the GL7.0 Global Land configuration (Walters et al., 2019). A number of changes have been introduced at RAL3, summarized in Table 2 (RAL Tickets #84 and #89). These attempt to minimise differences to the JULES global land settings, along with updates to albedo and transpiration. Many of the changes to the land surface inherited from GL7.0 and subsequent global UM science configurations are described by Williams et al. (2020), where they were introduced to target improved model drag. Although developed for global and therefore targeting generally coarser-scale model applications, there were no a-priori reasons to expect any dependence on model resolution.

Initial tests, however, found that the revised vegetation roughness length for momentum, $z_{0m}$, gave significant slowing of near-surface winds, with degradation to objective evaluation relative to observations. More acceptable performance could be obtained in tests across a number of domains by reducing $z_{0m}$ for the shorter vegetation tiles, such that RAL3 has $z_{0m}$ values of 0.1, 0.1 and 0.4 for C3 grass, C4 grass and shrubs respectively compared with GL7.0 values of 0.22, 0.22 and 1.0. Several factors may drive this requirement for retuning, including greater vegetation homogeneity in a grid area with higher resolution. A significant factor has been found to be the use of a much lower first vertical grid level in RAL3 (2.5m for winds, 5m for temperature, relative to 10m and 20m respectively in global model vertical level sets). The lowest grid level is coincident with the blending height for surface exchange so that smoother $z_{0m}$ are required in RAL3. In retuning $z_{0m}$ the ratio of scalar to momentum roughness lengths was also adjusted in order to keep the same roughness length for scalars as used in the GL7.0 configuration.

RAL3 uses the Met Office–Reading Urban Surface Exchange Scheme (MORUSES; Porson et al., 2010) with separate tiles for urban street canyons and roofs. This allows for varying building geometry at the grid-scale when calculating the

**Table 2.** Summary of differences between land surface settings in RAL3 and RAL2.

| Parameter/Process | RAL3 setting | RAL2 setting |
|---|---|---|
| 1. Roughness lengths for momentum $z_{0m}$ on vegetated tiles | Derived from FLUXNET observations in near-neutral conditions | Assumed a fixed fraction of the canopy height |
| 2. Sea surface drag and exchange | COARE algorithm for drag coefficient variation with wind speed, with cap on drag and reduction at high wind speeds (Gentile et al., 2021) | Fixed value of Charnock's coefficient, with cap on drag coefficient for winds above 30 m/s. |
| 3. Snowmelt | From underneath (on warm ground) and above | Occurs from above snow layers only |
| 4. Canopy snow storage | Stores on broad leaf and needle leaf trees | Storage on needle leaf trees only |
| 5. Canopy height ancillary | Derived from global forest canopy height LIDAR measurements (Simard et al., 2011) for trees, values reduced for grass tiles | Derived from land cover classification |
| 6. Soil hydraulics | Brooks (1965) to improve drainage into lower levels | van Genuchten (1980) |
| 7. Soil runoff generation | TOPMODEL (Beven and Kirkby, 1979) | PDM (Moore, 2007) |
| 8. Vertical gradient of soil suction | Assume linearity only for fractional saturation [l_dpsids_dsdz=$T$] | [l_dpsids_dsdz=$F$] |
| 9. Runoff from supersaturated soil | Excess water put in soil layer below. Excess from bottom soil layer becomes subsurface runoff | Excess water put in soil layer above. Excess from top layer becomes surface runoff |
| 10. Transpiration dependence on soil moisture | Allow plants to continue to transpire freely for soil moisture below critical point and linear reduction to zero below wilting point | Transpiration decreased by factor linearly reduced from 1 at critical point to 0 at wilting point |
| 11. Bare soil albedo | Include a zenith angle dependence that darkens the surface overall | |
| 12. Vegetation tile albedo | Derived from LOPEX93 (Hosgood et al., 1993) with separation between visible and near-infrared | |
| 13. Albedo calculation over sea | Inclusion of a chlorophyll climatology ancillary | |
| 14. Canopy radiation | Improved version of multi-layer radiation interception. Exponential decline of leaf nitrogen with canopy height proportional to leaf area index [can_rad_mod=6] | Multi-layer radiation interception. Exponential decline of leaf nitrogen through canopy [can_rad_mod=4] |

urban surface atmosphere scalar and momentum exchange. Currently MORUSES uses empirical functions to determine these morphology parameters based on the impervious sub-grid scale land cover fraction. These functions were originally based on high-resolution LIDAR data available for London (Bohnenstengel et al., 2011), developed for O(1km) grid lengths and are currently applied to urban areas in other regions and resolutions in the absence of suitable input data. Ongoing work across the Momentum Partnership aims to develop a more suitable dataset and workflow to incorporate newly emerging land cover data (e.g. Harper et al., 2023) and morphology datasets into updated and improved ancillary information. A correction to the MORUSES roof coupling was introduced in RAL3 to couple the roof radiatively to the underlying surface, reducing the diurnal temperature amplitude for roof tiles (RAL Ticket #74).

### 2.10 Lower boundary condition (ancillary files) and forcing data

In the UM/JULES, the characteristics of the lower boundary, the values of climatological fields and the distribution of natural and anthropogenic emissions are specified using ancillary files. Generation of ancillaries uses the Ancillary Tools and Suites (ANTS) libraries and the required ancillary science used with RAL3 is supported via a Regional Ancillary workflow. Table 3 lists the main ancillary dependencies for RAL3 and references to the source data from which they are created.

## 3 RAL3 revisions

Configuration updates have been delivered through three RAL3 revisions, described below. The initial release was re-labelled RAL3.0, and was only supported at UM versions 12.0, 12.2 and 13.0. Each subsequently released RAL3 revision then formed the standard supported definition for use across timescales, in preference to the previous revision. An exception arose, at time of writing, with both RAL3.2 and RAL3.3 continuing to be maintained and supported user options in light of more limited analysis of climate tests of RAL3.3.

### 3.1 RAL3.1

It was discovered that ’holes’ were apparent in vertical profiles of diagnosed reflectivity for extreme conditions in tropical systems during model spinup from its initial condition. Typically, where reflectivities at 1 km altitude were exceeding 55 dBz there would be a sudden reduction in simulated reflectivity. This was traced to the rain and snow coalescence leading to excessive change in droplet number concentration within a timestep when the concentration of condensed water approached 10 g/kg. The impact in the model was that the microphysics detected significant mass but no number and subsequently evaporated the rain. Later in the same timestep, the vapour was condensed to cloud before being converted to rain in following timesteps. The abrupt change from rain to cloud led to the formation of gaps or ’holes’ in reflectivity - referred to as ’radar holes’. To solve this issue, a CFL-like limit was added to number concentration process rates (RAL Ticket #375). In RAL3 this has been set to

limit the removal of number concentration to up to half of the existing number concentration within a timestep. RAL3.1 was supported at UM version 13.0, until superceded by the RAL3.2 revision.

### 3.2 RAL3.2

Unrealistically high surface temperatures could occur as a result of sudden very large increases in surface downward SW radiation, perhaps linked with rapid changes from very cloudy to clear sky, coinciding with a change from a stable to unstable BL. The buoyancy term used to determine the Monin-Obukhov length that is used in the calculation of the transfer coefficient for scalars is determined from the surface and atmospheric temperatures from the previous timestep. Under cloudy skies the air temperature can be warmer than the surface, implying a stable profile and small exchange coefficients. If on the next timestep the cloud clears, this leads to a large energy forcing term from the net surface shortwave that cannot be easily dissipated by the heat and moisture fluxes due to the very small transfer coefficient. On the following timestep the surface is identified as extremely unstable, with a large exchange coefficient, and the surface temperature then cools back down to realistic levels. Even so, this feedback can result in unphysically hot surface temperatures at selected points - referred to as 'hot spots'. RAL Ticket #311 was introduced to include a JULES code change that restructures the surface exchange code to update the buoyancy flux during the calculation of the surface transfer coefficients, removing the dependency on the previous timestep and so removing the hot spots. RAL3.2 continues to be supported at UM version 13.0 and 13.5.

### 3.3 RAL3.3

Assessment of pre-operational trials and subjective assessment of RAL3.2 for UK NWP highlighted a number of issues that were initially to be addressed through a dedicated package of changes as an NWP-specific branch of RAL3.2, but led to agreement of the definition of a further RAL3.3 revision, supported at UM version 13.5. This update includes a number of changes relative to RAL3.2 (RAL Ticket #649):

i An issue of particular concern to Operational Meteorologists arising through ongoing RAL3 assessment was of too little stratocumulus in winter anticyclonic conditions, including too rapid dissipation of low cloud. It was found that low cloud was sensitive to the monotonicity scheme for moisture advection (Section 2.4), and the issue could be mitigated by reverting settings for water vapour and potential temperature from the PMF scheme to tri-linear Lagrange (RAL Ticket #460). Whilst cloud amounts were increased, this change led to an increased cold bias in summer temperatures, and so could not be implemented in isolation.

ii Further radiation changes were therefore introduced to attempt to mitigate near-surface cold biases, with the scientific rationale being to make the radiation scheme more consistent with the use of CASIM microphysics (RAL Ticket #473). Specifically, ice particle optical properties were introduced, based on comparisons to estimates of short- and long-wave radiation from the Clouds and the Earth's Radiant Energy System (CERES) satellite-based instruments. These were

found to reduce low cloud amounts, leading to increased surface SW radiation and higher surface temperatures while not adversely impacting precipitation forecasts.

iii Two changes were introduced in CASIM to improve spin-up in space and time (RAL Ticket #479). Number concentrations for hydrometeors were set in lateral boundary regions, rather than set to zero, to improve spin up of precipitation into regional domains. Initialisation of hydrometeors to zero on the first timestep was also removed, to improve spin up in the first 1-2 hours.

iv It was also found that, while the Wilson and Ballard microphysics scheme could remove Murk aerosol from the atmosphere through being rained out, this representation was not present in the original CASIM implementation and so aerosol concentrations could accumulate unrealistically through a simulation, impacting visibility forecasts. A change was therefore introduced to enable washout of Murk aerosol within CASIM (RAL Ticket #471).

v Although changes introduced in RAL3.1 largely addressed the 'radar holes' issue, further evidence was reported of unrealistic discontinuities in simulated rainfall reflectivity. One cause was attributed to ice melting when unusual initial temperature structures were inherited from initial conditions, leading to evaporative cooling of snow and enabling snow to reach lower in the atmosphere, leading to readjustment of the particle size distribution with a small number of large droplets. To resolve this, a maximum size of snow before modification of the particle size distribution could take place was introduced (RAL Ticket #478). A second cause was related to size sorting of rain droplets, requiring introduction of an efficiency of rain collecting for larger mean droplet sizes, following Low and List (1982).

vi An error was discovered in the setting of one of the constants used in the wind gust diagnostic in computing the stability-dependent standard deviation of wind. Changing this parameter to the intended value reduces simulated wind gusts by around 10% (RAL Ticket #605).

vii The computational cost of RAL3 was also a consideration in governance of configuration definitions (see Section 6). Subsequent focus on optimising run time identified an increase in the time spent computing visibility diagnostics in RAL3. This was linked to unset values in the calculation of visibility in precipitation diagnostic when using CASIM that were previously set via the Wilson and Ballard scheme. Setting appropriate values led to tests running around 5%-10% faster with bit reproducible outputs (RAL Ticket #604).

## 4 Evaluation and Assessment of RAL3

Regional model development, evaluation and assessment is coordinated across the Momentum Partnership to ensure that the RAL configurations that underpin weather and climate research and applications are suitable and well characterized across a range of timescales, regions and resolutions. Figure 1 shows the model domains used in the evaluation of options for the RAL3 configuration. These domains vary in size based on location, model grid resolution, and application, and enable evaluation

across diverse climatic zones and for a range of weather phenomena of interest. The baseline configuration against which RAL3 was assessed was RAL2-M in the mid-latitudes and RAL2-T in the tropics and so the relative impact of RAL3 on results across the different domains is not directly comparable. Table A1 provides a summary of the simulation experiments from which results are presented in this paper, noting these do not represent the complete set of runs conducted to inform evaluation of RAL3 and candidate packages. Overall summary scores are presented below, with more detail for key weather variables in later sections 4.1-4.6. The impact of RAL3 on ensemble performance is discussed in Section 4.7 while feedback from targetted subjective assessment through daily forecast and evaluation in a number of Testbed activities with Operational Meteorologists are described in Section 4.8.

Where possible, a common Regional Model Evaluation and Development (RMED) Toolbox was used to provide consistent verification and diagnostic results of short-duration case study or cycling NWP experiments across multiple researchers and institutions. Summary scorecards in Figure 2 provide a snapshot of relative model performance across a range of weather variables for the initially released RAL3.0 configuration across a range on experiments in terms of the Ranked Probability Score (RPS), or Continuous Ranked Probability Score (CRPS), dependent on the variable of interest (Mittermaier, 2014). Figure 3 shows a summary of precipitation Fractions Skill Score (FSS; Roberts and Lean, 2008) for each experiment, computed by considering the fractional coverage of rainfall exceeding a given threshold (or percentile of a given distribution) within a certain spatial scale. Supplementary Figures S1 and S2 provide equivalent results based on assessments of the latest RAL3.3 revision. Upward pointing triangles (shaded green) indicate that the test model is performing better than the control and downward (purple) triangles indicate the test model is relatively worse in terms of the summary metric. The area of each triangle is proportional to the absolute improvement or deterioration of the score and triangles are outlined in black if the change is statistically significant at the 0.05 level using the Wilcoxon signed-rank test. Verification statistics are computed using the neighbourhood-based High Resolution Assessment framework (HiRA; Mittermaier, 2014) to reduce the influence of the 'double penalty' effect on results and their interpretation. A neighbourhood size of 7 grid lengths is used for consistency across assessments, corresponding to a spatial scale of 10-15 km for UK regional NWP tests and order 30 km for tropical case study testing over South East Asia for example. Overall, there is a general improvement in most variables for the verification scores across all regions and all revisions.

### 4.1 Precipitation and Convection

Multiple aspects of the simulation of precipitation are considered, including precipitation intensity (Section 4.1.1), the diurnal cycle (Section 4.1.2) and the interpretation of different evaluation metrics (Section 4.1.3).

#### 4.1.1 Precipitation intensity

New microphysics and cloud schemes in RAL3 have changed the precipitation characteristics for both mid-latitude and tropical domains compared with RAL2. Precipitation distributions (Figure 4) for RAL3.0 are in general improved compared to RAL2, with more light precipitation (less than 4 mm/h) and less heavy precipitation (in excess of 32 mm/h) leading to overall improved
agreement with radar or satellite-derived rainfall distributions. Supplementary Figure S2 shows consistent results for RAL3.3 for UK and Darwin domains. There is a continued over-estimation of more moderate precipitation (4 - 16 mm/h range), maintaining a wet bias overall when considering total rainfall accumulations. The overall improvement in precipitation intensity with RAL3 is evident for all domains and grid resolutions considered, including models with sub-km grid spacing (Figure 4(c)), and is consistent with results of the impact of the CASIM double-moment microphyics (Field et al., 2023). The fix for 'radar holes' introduced from revision RAL3.1 increased the most intense rainfall in tropical domains compared with the initial implementation, but distributions are still improved relative to RAL2 (Supplementary Figure S3).

RAL3 markedly improves the distribution of precipitation within convective cells compared to RAL2 for mid-latitude and tropical domains (Figure 5). For the UK, while the overall size of precipitation cells remains generally consistent between RAL2M and RAL3.0, the average intensity of each cell is reduced. Supplementary Figure S3 shows consistent results for equivalent assessments using RAL3.3. There is a small increase in the number of largest rain cells, indicating more organisation, with an increased average intensity of those cells. The change of precipitation organisation is particularly pronounced for the tropical Darwin domain, with similar results found for tests over Tropical Africa (not shown). While RAL2-T tends to simulate more intense circular cells without stratiform precipitation, RAL3 produces an increased number of relatively smaller cells with lighter rainfall intensity. There is also a considerable reduction in the number of most intense rainfall cells, with both changes improving the agreement to radar and GPM derived observations. These results imply an improvement in the simulation of convective events using RAL3, linked to weaker updraughts (not shown) and reduction in the strength of convection combined with more realistic cloud processes in both mid-latitude and tropical environments.

#### 4.1.2 Diurnal cycle of precipitation

There are notable improvements to the amplitude and timing of the diurnal cycle of convection in RAL3 (Figure 6). Results from 5-year free-running UK climate tests highlight a delay of the diurnal cycle of precipitation over the UK in summer by 2–3 h for all RAL3 revisions compared with RAL2-M, in better agreement with the radar-observed diurnal cycle. The peak in convection is relatively delayed to mid-to-late afternoon rather than midday as was found in RAL2-M. The improved timing is attributed, in part, to the removal of the stochastic boundary layer perturbations that were originally introduced in RAL1-M to help trigger convection earlier in the day. Earlier testing of RAL3 candidate packages without CASIM microphyics showed degraded model skill when the stochastic boundary layer perturbations were removed however. Microphysics therefore also plays some role in the timing of convective initiation and of increased precipitation. Figure 6(a) also illustrates an increase in magnitude of summertime rainfall in UK climate tests throughout the day, highlighting the overall remaining wet bias in RAL3 revisions.

The precipitation diurnal cycle timing in tropical domains is more consistent between RAL3 revisions and RAL2-T (Figure 6), although with evidence of improved amplitude for Darwin and tropical West Africa domains for example.

Subjective assessments (not shown) also indicate that the spin-up of small-scale convection from a 'cold start' (i.e. when initiated from global-scale model fields) in the tropics has improved. While small-scale showers took of order 12 h to spin up in simulations with RAL2-T, it takes around 9 h when using RAL3.

#### 4.1.3 Precipitation metrics

Verification of precipitation has been considered using two metrics: the RPS in the HiRA framework (Mittermaier, 2014) and the FSS (Roberts and Lean, 2008). These metrics are used to provide different information about model performance. The RPS focuses on binned distributions, calculating the differences between the bins in the model and observations to derive a measure of the differences in intensity of the precipitation across the entire distribution. However, slight differences across bin boundaries can penalise the model akin to the double penalty problem (e.g. if a simulated value of 0.21 mm falls in a different bin to the observation of 0.19 mm if a bin is defined at 0.2 mm threshold). In contrast, the FSS focuses on the location of rainfall events above a set threshold, or percentile, by converting precipitation to a binary field and giving all values above the threshold equal weight in deriving a summary score. Both metrics are subject to observational uncertainty regardless of the source of 'ground truth' - HiRA being typically applied using rain gauges and FSS comparing to radar.

Summary skill metrics for precipitation in tropical domains indicated beneficial impact of all RAL3 revisions relative to RAL2-T for both location and intensity (e.g. Figure 2(c)–(e) and Figure 3(c)–(e)). On the other hand, UK mid-latitude summary metrics indicate locational improvements (Figure 3(a),(b)) but degradations to the intensity (Figure 2(a),(b)) with respect to RAL2-M across all revisions (see also Supplementary Figure S1 and Figure S2). Investigations showed that the poorer intensity distributions relate to increased light rainfall values for 0.25 mm and 0.5 mm being above the observed frequency (e.g. Figure 4). This increased frequency implies greater areal extent (e.g. Figure 5) and thus will act in favour of the FSS due to more likely agreeing on spatial position at that threshold. Thus, extra detail is gained from consideration of both scores as opposed to one.

In addition to Testbed activities (Section 4.8), additional assessments were performed with key users (Operational Meteorologists and Hydrologists) to understand the extent to which objective scores agreed with their subjective experience of precipitation forecasts. Participants were asked to choose which configuration, if either, better represented precipitation compared to the radar for eight UK forecast cases (e.g. Figure 7). For the 4 cases for which RAL3.1 had better performance according to objective scores, this configuration was also preferred in the subjective assessment. For the 4 forecasts where objective scores suggested that RAL2-M was the better performing configuration, participants strongly preferred the RAL3.1 simulation for 2 cases and the response was more mixed for the other 2 cases (panel evenly split between RAL3.1, RAL2-M and neither solution). These results indicate that whilst there is an improvement to the precipitation in general with RAL3, research is required to relate the information provided through different objective scores to user experience, to help create more insightful evaluation for users of CP model predictions.

### 4.2 Lightning

Introduction of a new microphysics scheme required changes to the lightning diagnostic tuning parameters. These depend predominantly on the graupel and ice water paths produced by the model and can be tuned without affecting the model physics. Testing of the RAL3 candidate packages in simulations over India (e.g. Figure. 8) indicated that parameters $k1$ and $k2$ from the

McCaul et al. (2009) scheme needed to be increased by factors of five and three, respectively, to obtain comparable lightning rates to those using RAL2. These adapted parameter choices were found to produce similar results in comparison to RAL2 over the UK domain (not shown), and therefore considered suitable for use across different model domains. A further assessment and tuning of parameters with a UK focus is in progress ahead of operational UK NWP implementation of RAL3.3.

### 4.3 Clouds

Introduction of new large-scale cloud (Section 2.7) and microphysics (Section 2.6) have led to more realistic representation of cloud, consistent with the initial evaluation of the bi-modal cloud scheme (Section 2.7; Van Weverberg et al., 2021b, a). For example, Figure 2 indicates statistically significant improvements to summary verification scores for cloud cover and cloud base height across mid-latitude and tropical domains.

Mid-latitude results indicate relatively more medium and high cloud amounts in RAL3 (all revisions) than RAL2-M, and less low cloud at night. This is reflected in climatological cloud profiles from 5-year UK simulations (Figure 9), with the reduced frequency of low cloud also indicative of generally higher cloud base in RAL3. In the tropics, RAL3 has relatively less high cloud than RAL2-T, but considerably more medium and low cloud.

### 4.4 Temperature

Despite the improved simulation of clouds in RAL3, this did not directly translate into improved near-surface temperature results. Verification of simulations with the initial RAL3.0 release showed degraded temperature verification scores (Figure 2) linked with a generally cooler near-surface temperature and degradation of the cold bias that existed in RAL2. Subsequent revisions, in particular the 'radar holes' fix at RAL3.1 and use of more consistent radiation settings introduced in RAL3.3, have somewhat mitigated this and led to relatively improved temperature characteristics compared to RAL3.0 (e.g. Supplementary Figure S1). However the cold bias persists.

UK climate results highlight an enhanced diurnal temperature cycle in winter, and overall reduced temperatures in summer (Figure 9). The strong dependence on microphysics and cloud representation is illustrated by the jump in UK winter mean temperature at RAL3.1, while UK summer results are generally consistent across RAL3 revisions, although maximum temperatures are reduced for RAL3.3. Overall, Figure 10 shows a larger cold bias in winter mean temperature in RAL3 than for RAL2-M, consistent with the reduction in low cloud at night. Minimum temperatures (illustrated by the 1st percentile of coldest winter temperatures) are particularly cooler in RAL3 in northern UK areas, potentially linked to more lying snow cover. In summer, cooler temperatures reduce a warm bias in southeast England (Figure 11), but mean results are otherwise consistent between RAL2-M and RAL3 revisions.

### 4.5 Wind speed

Near-surface (10 m) wind speeds are generally slower in RAL3 (all revisions) relative to RAL2 across domains. Summary statistics (Figure 2) show a small improvement in wind speed verification over UK (e.g. reduced root mean squared error) in winter and summer assessment periods (though more neutral with RAL3.0 than for RAL3.3 in Supplementary Figure S1). RAL3 wind performance is notably improved for tropical domains, with reduced nighttime winds in particular reducing a strong wind bias (of order 1 m/s) found for RAL2-T. Case study simulations of extreme winds associated with tropical and extra-tropical cyclones also indicate weaker maximum winds using RAL3 relative to RAL2, with less intense depth of minimum storm pressure. The addition of frictional heating (Section 2.8) in RAL3 has helped to improve this given RAL2-T tended to over-deepen tropical cyclones. The general reduction of wind speed is consistent with the addition of sub-grid parametrized drag from orography, and increased surface roughness over vegetation as part of aligning global and regional model land settings.

### 4.6 Visibility

Verification of visibility relative to observations for UK and Darwin NWP tests show consistently improved performance of RAL3.0 (Figure 2). In contrast, Tropical Africa and South East Asia NWP results show a marked reduction in visibility forecast skill in RAL3.0 relative to RAL2-T. This is linked to the change from using a global constant value for aerosol concentrations in the visibility diagnostic to use of climatological aerosols with more regional variability (e.g. relatively increased aerosol over SE Asia region). In practice, use of a global constant in RAL2-T meant that while summary scores were apparently acceptable, the treatment of aerosol was inadequate as a forecasting tool (e.g. rarely predicting low visibility events). In RAL3, the move to more realistic geographical distributions has tended to increase both the probability of detection of lower visibility categories, but also the probability of false detection (e.g. now tending to predict low visibility events too frequently), noting also the relatively sparse network of observing stations.

Fog prediction remains particularly challenging but an important user application of convective-scale forecasts. Reduced wind speeds, and a reduction in the nocturnal low level cloud tending to reduce overnight temperatures in RAL3 (all revisions) has improved the representation of fog and low visibility over the UK in RAL3, particularly in winter and for the thickest fog and at later forecast lead times. Subjective assessments (e.g. see also Section 4.8) indicate that RAL3 forecasts still demonstrate too rapid development and tended to overestimate fog amounts. In contrast, subjective case study assessment of a number of fog events in the mid-latitude Melbourne and Perth NWP domains demonstrated relatively reduced fog fractions using RAL3.0, associated with increased dewpoint depression and reduced near-surface humidity.

Visibility in precipitation has also been improved in the RAL3 revisions, with visibility increased in precipitation areas relative to the RAL2 baseline. This is thought to be due to changes in the spatial density of rain droplets rather than to their size using the CASIM microphysics scheme. There is also an impact of predicting larger spatial extent of precipitation, for example during snow showers.
### 4.7 Ensemble performance

The evaluation of ensemble performance with RAL3 focuses on the growth of diversity between ensemble members (spread) and how it relates to forecast skill, noting that in an ideal unbiased ensemble with no observational error, the spread and error should be equal (e.g. Wilks (2011); Hopson (2014); Fortin et al. (2014)). The climatological ensemble variance should also lie within the observed climatological variance of the ensembles (Johnson and Bowler, 2009).

Results from a short 10-day 12-member ensemble trial focussed on Darwin (Figure 12) illustrate a marked improvement in ensemble characteristics using RAL3.0 for both near-surface temperature and 10 m wind speed, with an increase in the ensemble spread and decrease in the error. This improves a tendency for RAL-based ensembles to be over-confident.

More extensive ensemble trials focussed on the UK for month-long summer and winter periods, following the current operational UK ensemble (Porson et al., 2020), with 3 ensemble members run every hour on a 2.2 km horizontal grid to form an 18-member time-lagged ensemble over a six-hour period. Initial results showed an increase in the RAL3 ensemble spread for winter temperature relative to RAL2-M, but a decrease for the summer period. The spread of 10 m wind speed was also relatively reduced in both seasons using RAL3. There was no change to the spatial spread of precipitation (not shown), but improvement in the spatial skill for precipitation using RAL3, results in an improved spatial spread-error relationship (see Dey et al. (2014) for details on methodology).

To address the reduction in ensemble spread in the near-surface winds and summer temperatures, the perturbations applied to the model physics were reviewed. Model uncertainty is represented in the UK ensemble by the Random Parameter (RP) scheme (McCabe et al., 2016), where stochastic perturbations are applied to a sub-set of parameters in the boundary layer and microphysics schemes to reflect their uncertainty. In the initial ensemble trials, the RAL3 ensemble only had perturbations to the boundary layer parameters, with microphysics parameters perturbed in RAL2-M no longer active in RAL3. New parameters were subsequently introduced to the RP scheme for use with the bimodal cloud scheme and CASIM. These include parameters that control the fall speed for ice and snow, the mixed-phase overlap factor and the droplet number near the surface. The impact of these parameters on the RAL3 ensemble was found to be equivalent to the impact of the original microphysics parameters on the RAL2-M ensemble, resulting in only a small increase in screen level temperature spread for example.

Previous attempts to include land surface parameters in the RP scheme for RAL2-M had shown only limited impact, and therefore were not included in the current operational implementation. A new set of parameters related to roughness lengths for heat and momentum, leaf area index and surface albedo were tested again with RAL3, together with an additional parameter to perturb the coefficient for orographic form drag. The cumulative impact of new perturbed parameters shown in Figure 13 demonstrate very little impact on the ensemble error, but a substantial increase to the ensemble spread. This improves the spread-error relationships, particularly for summer near-surface temperature and for wind speed in both seasons. The ensemble spread using RAL3 over the UK is now comparable, or larger, than the ensemble spread using the previous RAL2-M baseline.

### 4.8 Testbeds and Subjective Assessments

Subjective evaluation of future forecast capabilities involving a diverse group of researchers and Operational Meteorologists has been enabled through design and delivery of Testbeds (e.g. Calhoun et al., 2021; Bain et al., 2022). Two potential options for RAL3 - a package including CASIM microphyics and an alternative candidate without, were assessed as part of a Winter 2021 UK Testbed. Daily 18-member ensemble forecasts over the UK, mirroring the MOGREPS-UK operational system (Porson et al., 2020), were run for a 4-week period and compared with a RAL2-M baseline. In general, the test configuration with CASIM (that was later implemented as RAL3) was preferred over the alternative, in particular due to the general increase in light rainfall. A number of less positive aspects were also noted, including development of cloud breaks under high pressure and warm sector conditions (later improved through RAL3.3 enhancements), and stronger and more rapid nighttime cooling and too early and spurious fog formation. Simulations with CASIM also led to more and larger clusters of showers with reduced precipitation intensities. An increase in low cloud ahead of frontal bands was noted, together with a tendency to trigger convection along those bands, and amplification of orographic enhancement in dynamic precipitation (considered to be excessive in some cases). During periods with snow, RAL3 tended to reduce the extent of convective snowfall over the sea, although this could not be verified, along with larger extent of snow showers over land.

During summer 2023, following release of the initial RAL3.0 configuration, RAL3 performance was assessed for daily 18-member ensemble forecasts over a region of south-west England with a grid-spacing of 300 m, termed the WMV, driven by operational MOGREPS-UK boundaries. This 13-week experiment built on the developments of Hanley and Lean (2024) focussed on a London domain, and was aligned with the Wessex Convection (WesCon) field campaign (P et al., 2021) with a strong focus on the challenge of predicting summertime convection. While the main focus was on assessment of the relative benefit of a higher resolution ensemble than available in operations, beneficial characteristics of RAL3 physics including improved organisation of convection into larger and less fragmented storms than MOGREPS-UK were highlighted.

A series of week-long Testbeds assessing RAL3 characteristics were also conducted with Operational Meteorologist in Malaysia, Indonesia and the Philippines during October 2021. The focus was on the spatial distribution, timing and intensity of precipitation, with each aspect given a subjective marking out of 9 (0-3 no/little skill; 3-6 some skill and 6-9 indicating skillful forecasts). Overall, both the RAL2-T baseline and RAL3 candidate performed well, with RAL3 found to perform better across metrics but with relatively small differences. Feedback in Malaysia shows improved timing of showers in RAL3, with later (improved) initiation and more accurate cessation in better agreement with GPM observations. Both configurations had too little and too slow propagation of precipitation from sea to land areas however.

These subjective assessments reinforce the broader evaluation results discussed in this paper and provide valuable insight on the characteristics of RAL3 performance and impact on decision-making in a forecasting context. This illustrates the value of routine monitoring and ongoing dialogue between model developers and users both well in advance of proposed changes, and long after implementation as part of ongoing Research-to-Operations and Operations-to-Research dialogue.

### 4.9 Implications for regional coupled atmosphere-ocean systems

With increasing interest in use of CP models as components of coupled environmental prediction frameworks (e.g. Castillo et al., 2022; Berthou et al., 2024), additional evaluation was conducted to assess the impact of RAL3 in the context of regional atmosphere-ocean coupled simulations. Note this took place after the initial release of RAL3, and therefore not part of the evidence considered as part of decision-making towards its definition. It is anticipated that understanding performance characteristics of regional coupled systems would become a core requirement in future RAL development cycles however.

Simulations of sea surface temperature (SST) using a km-scale coupled model (see Berthou et al. (2024) for experiment details) during marine heatwave conditions in June 2023 illustrate a tendency for upper ocean temperatures to be too cool when coupled to an atmosphere model using RAL2-M (Figure 14). A cool bias in excess of 1.5 K in the North Sea relative to the satellite-derived OSTIA SST (Donlon et al., 2012) is attributed to insufficient shortwave radiation in summer. RAL3 (all revisions) has more downwelling shortwave flux than RAL2-M, which led to warmer SST overall (Figure 14(b)). However, while coupled results with RAL3.2 maintained a relatively cooler shallow North Sea, temperatures were too warm by 0.5 to 1.5 K over much of the Northwest European shelf region surrounding the UK. Subsequent changes in RAL3.3 appear to better maintain cloud structures while adjustments to the radiation scheme to account for the new microphysics showed a largely beneficial impact on SST, and a reduction of domain average temperatures by around 0.7 K, in better agreement with observations.

## 5 Case Study assessment: 23 October 2022

Beyond the range of assessments presented in Section 4, RAL3 (and its subsequent revisions) have been evaluated through a number of case studies in different environments. Case study analysis allows deeper insight of the representation of physical processes and greater scientific assurance that differences in model characteristics are traceable to configuration changes.

As an example, a severe quasi-linear convective system (QLCS) associated with an occlusion which progressed from the south over the UK through the afternoon of 23 October 2022 (Kenneth et al., 2024: in review) is considered (Figure 15 and Supplementary Figure S4). Subjective comparisons of the outgoing longwave radiation and precipitation features indicated that case study simulations using RAL3 revisions and RAL2-M were broadly similar, and provided plausible forecasts of the case. The QLCS was in a similar position, being constrained by the same driving model, and with similar structures across forecasts, translating northwards across the UK.

All RAL3 revisions tend to have larger areas of stratiform rainfall, becoming more extensive with each revision, compared to RAL2-M. The focus for this case was a downburst located in south-east England (boxed area in Figure 15). Vertical cross-sections across this sub-region are considered to identify physical differences between the configurations and how this influences the convective activity. Potential vorticity (PV) cross-sections indicate broadly consistent large-scale conditions, with no distinct change in the predicted "weather story" across the revisions. Clear differences in average vertical cross-sections of both liquid and ice water content illustrate the impact of microphysics and cloud schemes introduced in RAL3. For RAL2-M

results, liquid water is more extensively transported aloft within updrafts, with impacts on latent heating and development of convection. Liquid water in RAL3 is relatively constrained to around 2.5 km altitude, but has much larger ice water content nearer cloud tops than RAL2-M. This implies ice processes to be relatively dominant in this case, leading to faster development of a more stratiform precipitation region and apprent improvement in the structure of the convection. Given the changes to the cloud ice and cloud water partitioning it can be inferred that the diabatic heating rates have changed (e.g. Flack et al., 2021) and thus there may be wider influences beyond impacts to the convection. Qualitatively, all RAL3 revisions have consistent characteristics and maintain common structural differences relative to RAL2-M. Variations in ice concentrations could help explain weaker convection in RAL3.1 and RAL3.2 (see Supplementary Figure S4). For this case, the correction to address the 'radar holes' issue between RAL3.0 and RAL3.1 results in more substantial quantitative differences to the maximum simulated reflectivity and ice water content than increments between other RAL3 revisions (Supplementary Figure S4). This result implies the 'radar holes' was not just limited to the tropics and that process-based analysis is a critical part of the model development process to detect potential issues.

## 6 Computational Performance

Enhancing simulation fidelity while maintaining sustainable computational costs is a key requirement for model development with both time-bound operational NWP and throughput-dependent production climate applications in view. A number of computational performance tests were conducted on the Met Office Cray XC40 supercomputer for a test domain with 1.5 km grid spacing focussed on south-west UK and with a 4.4 km grid over Singapore (Table 4). Average simulation times, and the relative costs of different RAL configurations are dependent on the choice of compiler optimisation level, with "fast" typically used when RAL has been used in operational NWP, while "safe" had tended to be used for research and production climate applications.

All timings quoted in Table 4 are based on UM version 13.5 tests, with RAL3.2 results representative of the relative cost difference between RAL2 and the initial RAL3.0, RAL3.1 and RAL3.2 implementations at UM version 13.0. Initially, RAL3 was order 35-50% more expensive than RAL2 with "safe" compiler optimisation settings, and 30-35% with "fast" settings. These considerable increases were attributable to introduction of the more complex double-moment CASIM scheme. The cost of moisture advection is also increased due to advection of additional variables used by CASIM and the addition of the "fountain buster". The RAL Governance Group, in particular NWP and climate users, considered the increased cost of RAL3 to be acceptable in light of enabling the benefits of improved model performance. However, work is ongoing to further analyse and find optimisations. This led to subsequent improvement to model cost of around 6% for UK tests and 13% for Singapore-focussed tests through introduction of RAL Ticket #604 into RAL3.3 (also implemented for the RAL3.2 revision supported at UM13.5).

## 7 Discussion and Conclusions

Release of a globally applicable definition of the Met Office Unified Model–JULES Regional Atmosphere and Land configuration that performs effectively has been a long term goal across the Momentum Partnership. The RAL3 configuration uses the same dynamical core as the previous RAL1 (Bush et al., 2020) and RAL2 (Bush et al., 2023) configurations, but with introduction of a fountain buster scheme to correct locally convergent flow. A large advance compared to RAL1 and RAL2 is the unification of the tropical and mid-latitude configurations. This unification was achieved in part through the introduction of two new parametrizations for microphysics and cloud: CASIM (Field et al., 2023) and a bi-modal cloud scheme (Van Weverberg et al., 2021b, a) which replace the different schemes used in the mid-latitudes and tropics in RAL2. Other factors important to delivering a unified configuration include changes to the boundary layer turbulent mixing, updated land parametrization and parameters to be more consistent with those used in the global configuration, and a review of other differences between the previous mid-latitude and tropics optimisations (such as removal of stochastic boundary-layer perturbations that were only active for mid-latitude domains).

The RAL3 development and evaluation process has implemented the recommendations advocated by Bush et al. (2023). For example, multi-annual climate testing over UK and Africa domains were introduced as a core part of the assessment of package options and decisions on the configuration definition. While it has not been practical to test the impact of all proposed science changes at a package level using longer simulations, the assessment of headline package options provided assurance on performance characteristics ahead of RAL3 release for use in research and production climate applications. Future evolution of this approach might include climate ensemble testing, either to understand long-term characteristics of perturbation methods, or sensitivity of results to different driving models or reanalysis.

Ensemble-focussed assessment occurred earlier in the RAL3 development than previously. However, the need to embed definitions of random parameters and other stochastic aspects within new physics much earlier in the process has been highlighted (Section 4.7). The earlier use of ensembles requires flexible ensemble tools and experiments, including design of cost-effective but insightful testing strategies using combinations of reduced model resolution, smaller number of members, simple ensembles (e.g. Flack et al., 2019) and simplified cycling strategies.

The introduction of regional coupled testing and evaluation ahead of configuration release (Section 4.9) is recommended for future development cycles given increased user applications.

With increasing complexity of regional models, particularly with proposed introduction of coupled testing, process-based evaluation becomes more important (Section 5). This importance arises due to the need of users to understand the impact of the changes on fundamental weather processes so they retain their trust in the configuration and can detect the improvements in its use (e.g. when forecasting or considering changes in weather extremes). Process-based evaluation (Section 5) is therefore recommended to be a more prominent feature of decision-making in future RAL cycles.

Whilst the RAL3 evaluation has been more comprehensive than for previous development cycles, with broad involvement across the Momentum Partnership, it was not able to capture all weather types due to the testing periods considered. Further evaluation, including towards operational implementations across the Momentum Partnership, sampled different weather con-

ditions to the evaluation phase, and identified issues that would cause negative impacts for users that would have otherwise been missed. This highlights the need for a greater variety of evaluation periods and targetting of extreme events. Following the initial release of RAL3 in summer 2022, since re-named as RAL3.0, priority issues have been investigated and addressed through definition of successive RAL3 revisions, with the development process overseen by a RAL Governance Group. Differences to the configuration definition, and resulting simulation characteristics, between RAL3.0 and subsequent revisions (RAL3.1, RAL3.2 and RAL3.3) remain small relative to the physics and performance changes between RAL3 and relevant RAL2 baselines. RAL3.2 and RAL3.3 are supported for operational and research use across weather and climate timescales.

### 7.1 Characteristics of RAL3

The RAL3 configuration has many changed characteristics compared to relevant RAL2-M (mid-latitude) and RAL2-T (tropical domain) baselines. These are summarized below, with reference to specific RAL3 revisions where relevant.

i There is less "heavy" precipitation and more "light" precipitation. From RAL3.1 onwards the heavy precipitation increased but still represents an improvement on RAL2. The overall wet bias persists.

ii Cloud bases and fractions have improved across RAL3.0 and 3.3. There was a slight degradation in RAL3.1, and minimal changes at RAL3.2. Low cloud cover has improved in anticyclonic regimes in RAL3.3 and represents an improvement on RAL3.2, and RAL2.

iii Visibility has improved, particularly in the tropics. In the mid-latitudes fog density and visibility in precipitation has improved. However, in revisions before RAL3.3 the fog dissipated too quickly. This dissipation rate has improved to some extent in RAL3.3. However, further investigation is required to determine how much of an improvement the slower dissipation of fog is relative to RAL2.

iv In all RAL3 revisions the convection appears to have improved structure with a clear stratiform region. However, the convection was successively weaker in RAL3.1 and RAL3.2 revisions. In RAL3.3 convection is at a similar strength to RAL3.0, which was reduced compared to RAL2. The strength of convection is improved in RAL3 given that convection was too strong in RAL2.

v A pre-existing cold temperature bias in the mid-latitudes was degraded in RAL3.0, RAL3.1 and RAL3.2. However, since RAL3.3 the cold temperatures have relatively improved to be comparable RAL2, persisting the previous cold temperature bias. Maximum temperatures in RAL3.3 are reduced for all regions assessed.

vi Surface winds have been consistently represented across all revisions. There is a general reduction in the surface winds (reducing slightly with each successive revision). This is beneficial during nocturnal hours and has mixed results during daytime hours.

vii Ensemble spread was initially lower (RAL3.0) during the evaluation phase. However, since the update to the random parameter scheme the ensemble spread has increased, and the overall spread-skill relation is improved.

Despite some degradations, the overall result is that RAL3 delivers a major improvement upon RAL2, regardless of revision considered. The RAL3.3 revision is recommended for use across research, operational NWP and production climate applications.

### 7.2 Applications and Operational Implementation of RAL3

The evaluation of RAL3 continues through a range of research applications and ongoing subjective evaluation to deepen the understanding of RAL3 characteristics. These activities also inform ongoing development priorities for future RAL cycles. The range of research using the RAL3 configuration across the Momentum Partnership and academic users illustrates the value of a unified definition across domains, spatial and temporal scales. For example, Jones et al. (2023) used RAL3.0 in tropics-wide 2.2 km resolution experiments to gain insight on the impact of domain size on tropical precipitation. Maybee et al. (2024) have examined the response of Mesoscale Convective Systems in West Africa to moisture and wind shear, finding that RAL3.2 can capture observed relationships more strongly than in previous CP simulations. Senior et al. (2023) demonstrated reduced rainfall intensities in RAL3.0 (relative to RAL1-T) in better agreement with observations overall and provided clearer indications between extreme precipitation days and others over Sumatra. Hanley and Lean (2024) further demonstrated the performance of RAL3.1 when applied in an urban-scale model ensemble focussed on London with 300 m grid spacing, while Hall et al. (2024) assessed surface temperature predictions using RAL3.3 down to 100 m grid spacing compared with Landsat observations. RAL3.2 is being used extensively in a number of convective-scale regional climate experiments. These experiments include an ongoing pan-Africa 4.4 km ensemble as a major enhancement to the first CP4-Africa simulations (Stratton et al., 2018).

The RAL3.2 configuration is now in operational use at both NCMRWF (India) and MSS (Singapore). At MSS, full-cycling data assimilation hindcast trials were conducted to assess the impact of RAL3.2 (SINGV-DA vn6.0, 1.5 km grid) relative to the baseline operational SINGV-DA (RAL1-T). In general, there were large improvements in the precipitation forecasts over the whole regional domain, particularly for light rain. Over Singapore itself, the rainfall forecasts were less biased, with higher probability of detection, particularly for light rainfall events. Wind forecasts over Singapore were marginally improved, with reduced errors and smaller biases, attributed to surface drag changes. Since operational implementation of RAL3.2, positive feedback has been received from MSS forecasters, along with the historical record for the SINGV summary skill index being exceeded, providing early indications of improved forecasts and user benefits.

The introduction of RAL3.0, and subsequently RAL3.2, at NCMWRF has improved the spatial distribution of precipitation in the 4.4 km grid NCUM-R operational regional model (e.g. Niranjan Kumar et al., 2023). Assessment of 45 cases during the summer 2022 monsoon period showed improved FSS for both precipitation and lightning (except for low lightning flash rates of less than 5 per day) using RAL3 relative to the previous operational RAL2-T baseline. NCMRWF have also coupled the UK Chemistry and Aerosol (UKCA) scheme to RAL3.2 in the 300 m grid DM-Chem urban-scale application over Delhi (Jayakumar et al., 2021; Gordon et al., 2023), to provide enhanced air quality and fog prediction capability. The DM-Chem model with RAL3.2 has also been used to support process-based research into the impact of urban and aerosol processes on fog development over the Indo-Gangetic Plain (Anurose et al., 2024).

Other Momentum Partners, including the Met Office (UK), Bureau of Meteorology (Australia), and NIWA (New Zealand), are working towards implementation of RAL3 within upcoming operational NWP upgrade cycles across ensemble and deterministic applications across spatial scales. Development of a pan-Australia convective-scale capability at the Bureau further illustrates the utility of the unified RAL3 configuration valid for both tropical and mid-latitude regions. RAL3 revisions have also been successfully applied on a global grid with km-scale spacing at the Met Office, in support of K-Scale research (Jones et al., 2024). These applications highlight the increasing appetite to further align parametrizations previously optimized for either coarse-scale global or convective-scale regional applications, with research progressing towards scale-aware schemes. More immediately, RAL3 sets the basis for a transition to the next-generation LFRic atmosphere model code under development across the Momentum Partnership. This development will ensure the benefits of regional models at CP scales can continue to be realized across a range of domains, space and timescales on future computing architectures, underpinning provision of weather and climate services for society.

*Code availability.*

*Obtaining the Unified Model.* The Met Office Unified Model (UM) is available for use under a closed licence agreement. A number of research organizations and national meteorological services use the UM in collaboration with the Met Office to undertake research, produce forecasts, develop the UM code, and build and evaluate models. For further information on how to apply for a licence, please contact scientific_partnerships@metoffice.gov.uk. See also http://www.metoffice.gov.uk/research/modelling-systems/unified-model (last access: 14 June 2024). UM documentation papers are accessible to registered users at https://code.metoffice.gov.uk/doc/um/latest/umdp.html (last access: 14 June 2024).

*Obtaining JULES.* The Joint UK Land Environment Simulator (JULES) is freely available to any researcher for non-commercial use. Further information on requesting access and the JULES Terms and Conditions are accessible via http://jules-lsm.github.io/access_req/JULES_access.html (last access: 14 June 2024). The JULES user manual is available at https://jules-lsm.github.io/ (last access: 14 June 2024).

*Obtaining the Flexible Configuration Management system.* The UM and JULES codes were built using the fcm_make extract and build system provided within the Flexible Configuration Management (FCM) tools. UM and JULES codes and Rose suites were also configuration managed using this system. FCM releases can be obtained via the Github repository https://github.com/metomi/fcm/releases (last access: 14 June 2024), under GNU General Public License. Further information and user documentation is provided at http://metomi.github.io/fcm/doc/user_guide/ (last access: 14 June 2024).

*Obtaining Rose and Cylc.* The Rose framework was used for defining UM/JULES workflows. This is free software available under GNU General Public License. Further details are available at https://github.com/metomi/rose (last access: 14 June 2024). Cylc is a general purpose workflow engine that manages and runs cycling systems including UM/JULES workflows. This is available under GNU General Public License. Further details are available at https://cylc.github.io (last access: 14 June 2024) and Oliver et al. (2019). *Obtaining RAL3 workflows and configuration* Workflows used in development of RAL3 are available to any licensed user of both the UM and JULES via the Met Office Science Repository Service (MOSRS) via https://code.metoffice.gov.uk/trac/roses-u/ (last access: 14 June 2024). Further support for using MOSRS are provided at https://code.metoffice.gov.uk/trac/home (last access: 14 June 2024). Details of RAL3 configuration parameters are shared through the zeonodo asset https://doi.org/10.5281/zenodo.13957006 (last access:20 October 2024) registered to support documentation of this paper.

## Appendix A: Appendix A: Simulation experiment details

Table A1 summarises information on simulations discussed in this paper. Each experiment is listed with an identifier that corresponds to the workflow that was used with rose/cylc frameworks to define and manage the execution of case study, NWP trial or climate simulations. Workflows are archived and revision-controlled using the Met Office Science Repository Service, and contain the information required to extract and build the code as well as configure and run the simulations. Workflows used in development of RAL3 are available to any licensed user of both the UM and JULES.

*Author contributions.*

MBu led the overall RAL3 development and evaluation process. The manuscript has been prepared with contributions and review from all co-authors, with drafting led by MBu, DLAFl, HWLe, SIBo and CFr. Code owners and developers of new schemes and modifications introduced into RAL3 include APLo and JEd (Boundary Layer), MBe (Land Surface), PFi and AHi (CASIM microphysics), KVWe (Bimodal cloud), JWi and SMoh (Lightning), with additional parametrization contributions from IBo, KFu, MAHe, CMo, SVo, MWh and KDWi. RAL3 evaluation studies described in the paper were conducted with technical and scientific contributions from all co-authors, with a variety of Momentum Partner activities coordinated by MBu (Met Office), CFr (Bureau), HZh (MSS), SMoh (NCMRWF), SMoo (NIWA). Ensemble developments and evaluation were led by AMc. Testing and evalutation with a focus on UK NWP, including subjective assessment activities, were conducted and assessed with contributions from DLAFl, JKBr, SeCo, GDo, AFi, RNo, APo, NRo, DWa, SWe and MWe. UK-focussed sub-km testing was conducted by KHa and HuLe. Australia-focussed NWP tests were conducted by APLo, CFr, ShCo, SRe, BRo, C-HS. South-east Asia focussed NWP tests were conducted by RWJo, JCKLe, CSa and HZh. Africa-focussed NWP tests were conducted by JWa. India-focussed NWP tests were conducted by AJa and SMoh. Short climate tests were conducted and analysed by CJSh, KHa, and STu. Regional coupled tests were run and assessed by SBe. The UK case study assessment was led by DLAFl.

*Competing interests.*

The authors declare that they have no conflict of interest.

*Acknowledgements.* The development and assessment of the Regional Atmosphere Land configuration is possible only through the contributions of a large number of people that exceeds the list of authors of this paper. We would particularly wish to acknowledge the underpinning development and maintenance of the technical tools that support this endeavour, notably all code developers of the Unified Model and JULES, and those who support use of tools and workflows to run simulation experiments and analyse their outputs.

The GPM IMERG Late Precipitation L3 half-hourly 0.1 degree x 0.1 degree V06B and V07B data were provided by the NASA/Goddard Space Flight Center's Goddard Earth Sciences Data and Information Services Center and PPS, which develop and compute the GPM IMERG Late Precipitation L3 Half Hourly 0.1 degree x 0.1 degree as a contribution to GPM, and archived at the NASA GES DISC.

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

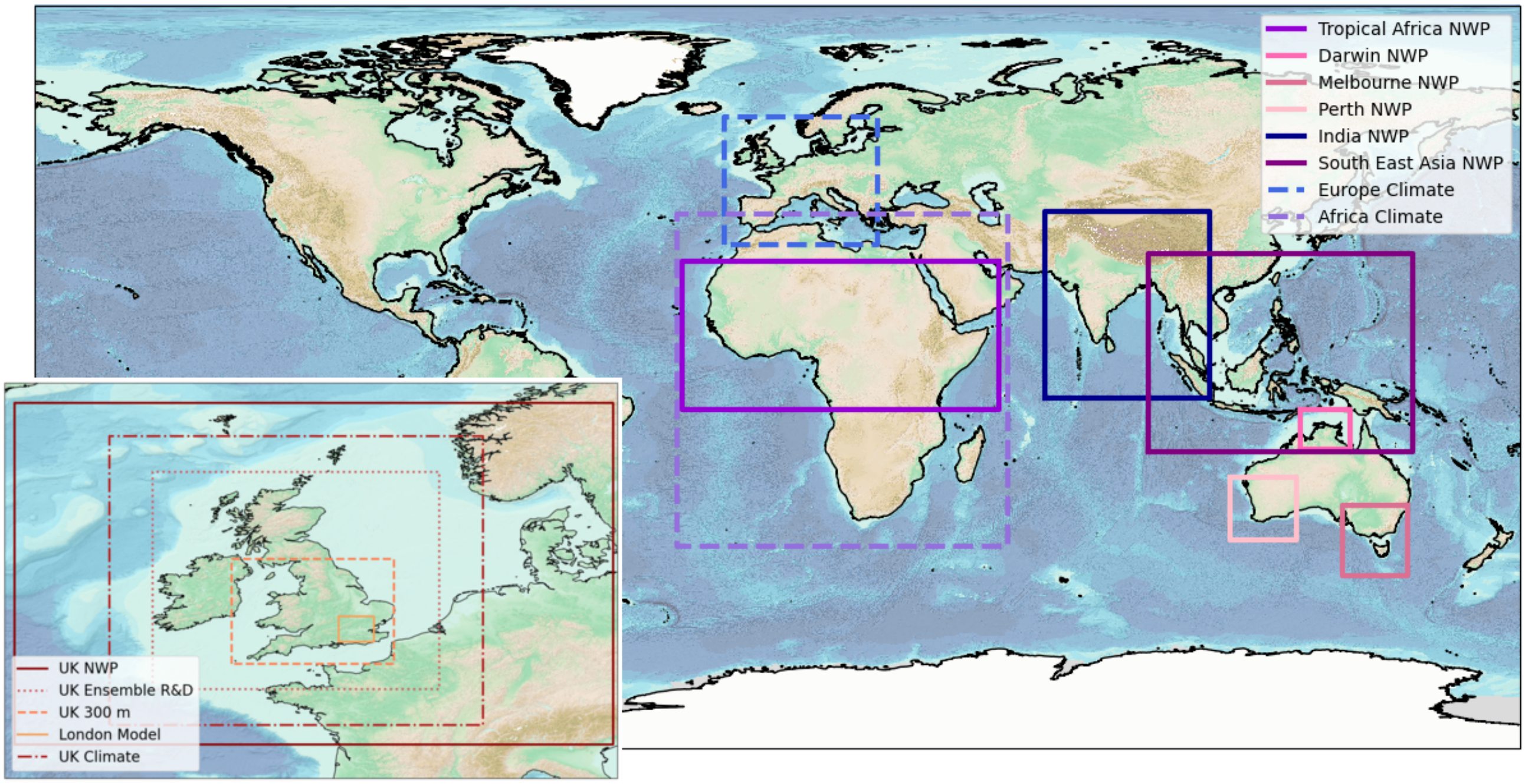

**Figure 1.** Regional domains considered within the RAL3 evaluation process across timescales. UK-focussed domains are highlighted in the inset map.

**Table 3.** Standard ancillary requirements and default source datasets used to specify inputs to RAL3 applications. The following acronyms for sources of data are used: CCI - European Space Agency Climate Change Initiative; IGBP - International Geosphere-Biosphere Programme; SRTM - Shuttle Radar Topography Mission; GLOBE - Global Land One-kilometer Base Elevation; ITE - Institute of Terrestrial Ecology; HWSD - Harmonized World Soil Database; MODIS - Moderate Resolution Imaging Spectroradiometer; ICESat - Ice, Cloud and land Elevation Satellite; OSTIA - Operational Sea Surface Temperature and Ice Analysis; NCDC - National Climatic Data Center; NAEI - National Atmospheric Emissions Inventory; EMEP - European Monitoring and Evaluation Programme

| Ancillary field | Source data | Source grid | References | Notes |
|---|---|---|---|---|
| Land/Sea mask | CCI v1 | 330 m | Hartley et al. (2017); Harper et al. (2023) | Applied worldwide. |
| | IGBP v2 | 1 km | Loveland et al. (2000) | Applied for UK NWP only |
| Orography | SRTM | 100 m | USGS | Only available to 60 $^o$N and 56 $^o$S. |
| | GLOBE | 1 km | Hastings et al. (1999) | |
| Topographic index | HydroSHEDS | 450 m | Marthews et al. (2015) | New requirement for RAL3 due to inclusion of TOPMODEL. |
| Land use and cover | CCI v1 | 330 m | Hartley et al. (2017); Harper et al. (2023) | Applied worldwide. |
| | ITE Land Classification | 1 km | Bunce et al. (1996) | Available over Great Britain only |
| | IGBP v2 | 1 km | Loveland et al. (2000) | Applied over non-GB areas of UK NWP regions only |
| Soil properties | HWSD v2 | 1 km | FAO and IIASA (2023) | |
| Leaf area index | MODIS collection 5 | 4 km | e.g. De Kauwe et al. (2011) | Mapped to 5 plant types |
| Plant canopy height | IGBP v2 | 1 km | Loveland et al. (2000) | Derived from land use and mapped to 5 plant types |
| | ICESat | 1 km | Simard et al. (2011) | Global tree canopy height |
| Bare soil albedo | MODIS collection 4 | 0.05$^o$ | Houldcroft et al. (2009) | |
| SST and sea ice | OSTIA | 0.05$^o$ | Donlon et al. (2012) | e.g. NWP case studies |
| | US NCDC SST climatology | 1$^o$ | Reynolds et al. (2002) | e.g. climate simulations |
| | HadISST 1961-1990 climatology | 1$^o$ | Rayner et al. (2003) | Climatological background |
| Ozone | UGAMP ozone climatology | 2.5$^o$ | Li and Shine (1995) | |
| Murk aerosol | UK NAEI, ENTEC ship emissions and EMEP emission inventories | Point sources | | |
| Aerosol climatology | CLASSIC for NWP, EasyAerosol for climate | | Walters et al. (2019) Voigt et al. (2014) | |
**Table 4.** Relative cost of average run times (based on 12 sets of 3-hr simulations) for RAL3.2 and RAL3.3 revisions supported at UM version 13.5, as % change relative to relevant RAL2-M (UK) and RAL2-T (Singapore) baselines.

| Domain | Compiler option | RAL3.2 | RAL3.2+#604 | RAL3.3 |
|---|---|---|---|---|
| Singapore domain (4.4 km) | Safe | +49% | +36% | +39% |
| Singapore domain (4.4 km) | Fast | +34% | +22% | +25% |
| South-west UK domain (1.5 km) | Safe | +37% | +32% | +33% |
| South-west UK domain (1.5 km) | Fast | +31% | +25% | +26% |

**Table A1.** Simulation details of experiments conducted to support evaluation of RAL3 discussed in this paper across timescales, resolutions and domains of interest. This covers a sub-set of domains and experiments used through the RAL3 development to assess aspects of candidate configuration packages, initial release version and subsequent revisions. Simulation workflows, with 'mi-' or 'u-' identifiers are archived on the Met Office Science Repository Service https://code.metoffice.gov.uk/trac/home

| Experiment | Description | RAL3 | RAL2 |
|---|---|---|---|
| UK Climate | Free-running 6-year simulations, initialised 01/01/2007. 2.2 km model over UK with 60 s timestep, nested within UKCP18 12 km RCM spanning Europe, driven by ERA-Interim reanalyses and SST derived from observational analysis (Reynolds et al., 2002). First year of simulations is discounted from analyses as spin-up. | mi-bd046 | mi-bc053 |
| UK NWP winter 2022 | Deterministic 6-hourly cycling data assimilation NWP trial, using UKV model domain with variable resolution and 2.2 km grid spacing in inner domain (Tang et al., 2013), nested within operational global NWP. Testing RAL3 revisions. Trial period 1 December 2021 - 28 January 2022 (59 days). | mi-bf303 | mi-be497 |
| UK NWP summer 2022 | As UK NWP winter trial. Testing RAL3 revisions. Trial period 8 July - 17 August 2022 (41 days). | mi-bf312 | mi-bf338 |
| UK NWP winter 2020 | As UK NWP winter trial. Testing initial RAL3 release. Trial period 2 December 2019 - 22 January 2020 (52 days). | mi-bc895 | mi-bb676 |
| UK NWP summer 2019 | As UK NWP winter trial. Testing initial RAL3 release. Trial period 16 June - 4 August 2019 (50 days). | mi-bc924 | mi-bb692 |
| Darwin NWP | Deterministic 12-hourly cycling trial with 1.5 km grid spacing, initialised from ECMWF operational analysis and driven by ECMWF global forecast lateral boundary conditions out to. Trial period 21 January - 19 March 2017 (58 days) | u-co345; u-da769 | u-cj172 |
| Tropical Africa NWP | Deterministic 12-hourly cycling trial with 4.4 km grid spacing run at 06Z and 18Z daily out to 72h. Nested within 17 km (GA7.2) operational MetUM global NWP. Trial period 1 August - 14 September 2020 (45 days) | u-ci247 | u-ce073 |
| SE Asia NWP | Deterministic 12-hourly cycling trial with 4.4 km grid spacing run at 00Z and 12Z daily out to T+120h. Nested within 17 km (GA7.2) operational MetUM global NWP. Trial period 1 January - 30 January 2020 (30 days). Only 00Z results analysed here. | u-ci088 | u-ci088 |
| India NWP | Deterministic 4.4 km NCUM-R daily run simulations with 90 vertical levels. Experiment period March - June 2019 | u-ck361 | u-cl425 |
| UK ensemble winter | Month-long 18-member trials using MOGREPS-UK model domain with variable resolution and 2.2 km grid spacing in inner domain (Porson et al., 2020), nested within operational global MOGREPS-G ensemble. RAL2-M driven by GC4, RAL3.3 with/without new random parameters driven by GC5. Trial period 1 December 2021 - 1 January 2022 | mi-bf883 mi-bf597 | mi-be989 |
| UK ensemble summer | As UK NWP ensemble trial. Trial period 15 July - 15 August 2022 | mi-bg042 mi-bf700 | mi-bf739 |
| Darwin ensemble | 10-day 12-member trials comparing RAL3.0 with RAL2-T, initialised every 6 hr using 2.2 km grid spacing. Trial period 29 November - 8 December 2020 | | |
| UK summer case study | Deterministic case study simulations over UK for summer 2021, using 1.5 km and 300 m grid spacing models. | u-cj967 | u-ce890 |

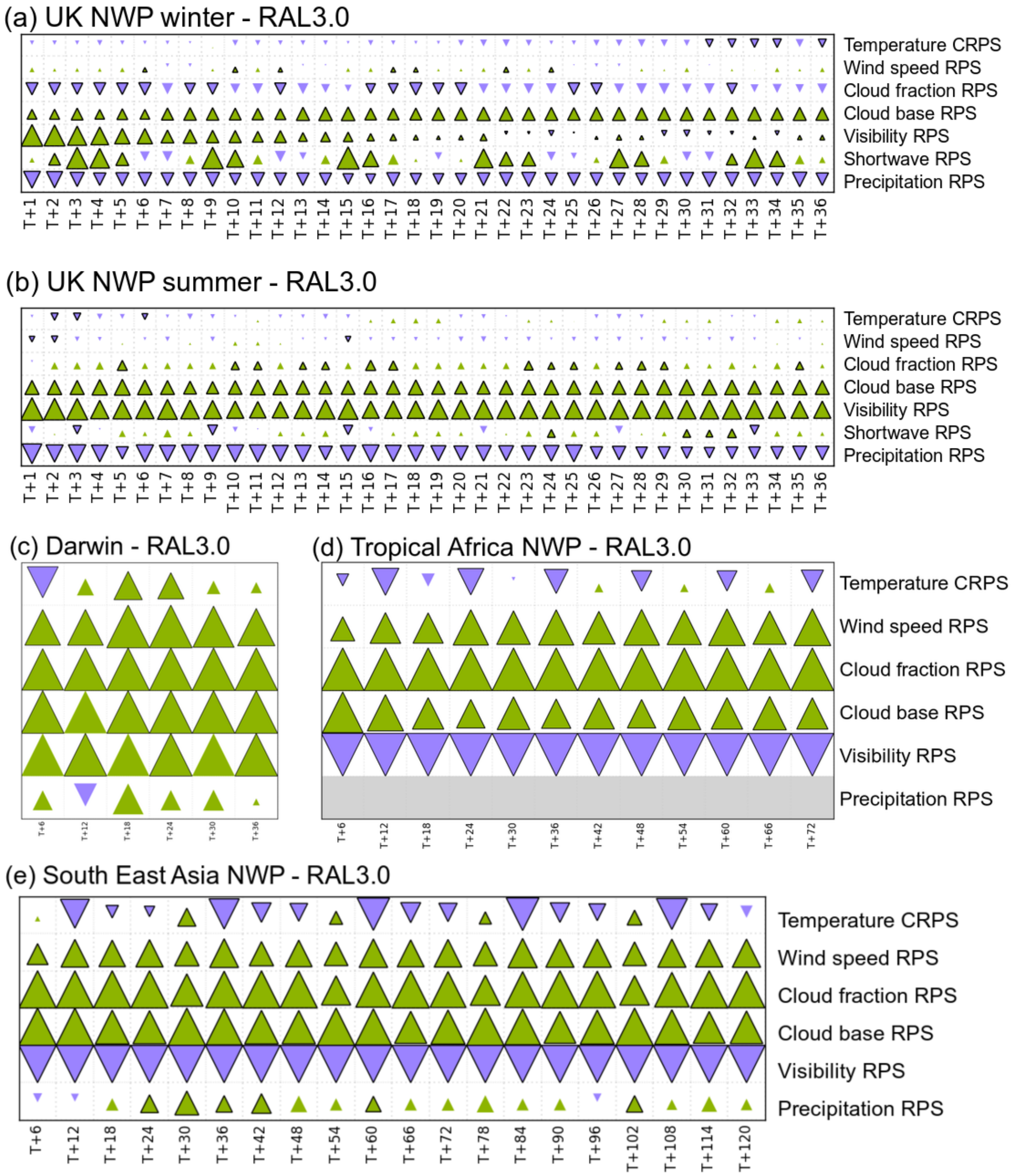

**Figure 2.** Summary scorecards for the percentage differences between the initially released RAL3.0 configuration and region-dependent RAL2 baseline for the Ranked Probability Score or Continous Ranked Probability Score (dependent upon variable) calculated within the High Resolution Assessment (HiRA) framework for a neighbourhood of seven grid lengths from NWP simulation experiments for a) UK winter [to T+36h, hourly statistics, RAL3.0 vs RAL2-M], b) UK summer [to T+36h, hourly, RAL3.0 vs RAL2-M], c) Darwin, Australia [to T+36h, 6-hourly, RAL3.0 vs RAL2-T] d) Tropical Africa [to T+72h, 6-hourly, RAL3.0 vs RAL2-T] and e) South-East Asia domains [to T+120h, 6-hourly, RAL3.0 vs RAL2-T]. Green triangles represent an improved score, and purple triangles represent a degradation in the score. Black borders mark changes that are statistically significant at a 95% significance level.

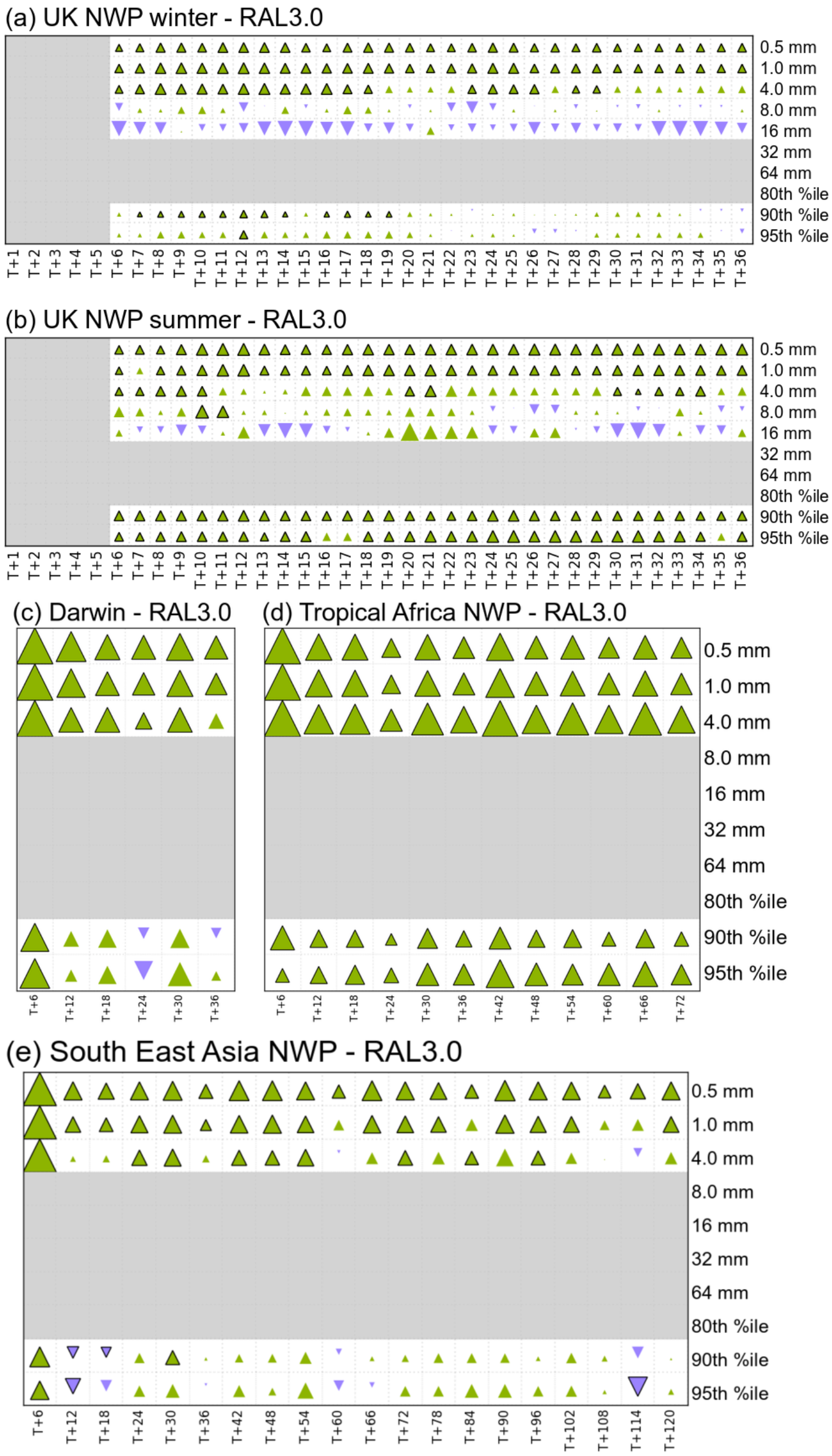

**Figure 3.** Summary scorecards for the percentage differences between the initially released RAL3.0 configuration and region-dependent RAL2 baseline for the Fractions Skill Score (FSS) at differing thresholds (both absolute and percentile). The FSS presented are for neighbourhood widths of five grid lengths from NWP simulation experiments for a) UK winter [to T+36h, hourly statistics, RAL3.0 vs RAL2-M], b) UK summer [to T+36h, hourly, RAL3.0 vs RAL2-M], c) Darwin, Australia [to T+36h, 6-hourly, RAL3.0 vs RAL2-T], d) Tropical Africa [to T+72h, 6-hourly, RAL3.0 vs RAL2-T] and e) South-East Asia domains [to T+120h, 6-hourly, RAL3.0 vs RAL2-T]. Green triangles represent an improved score, purple triangles represent a degradation in the score. Black borders mark changes that are statistically significant at a 95% significance level.

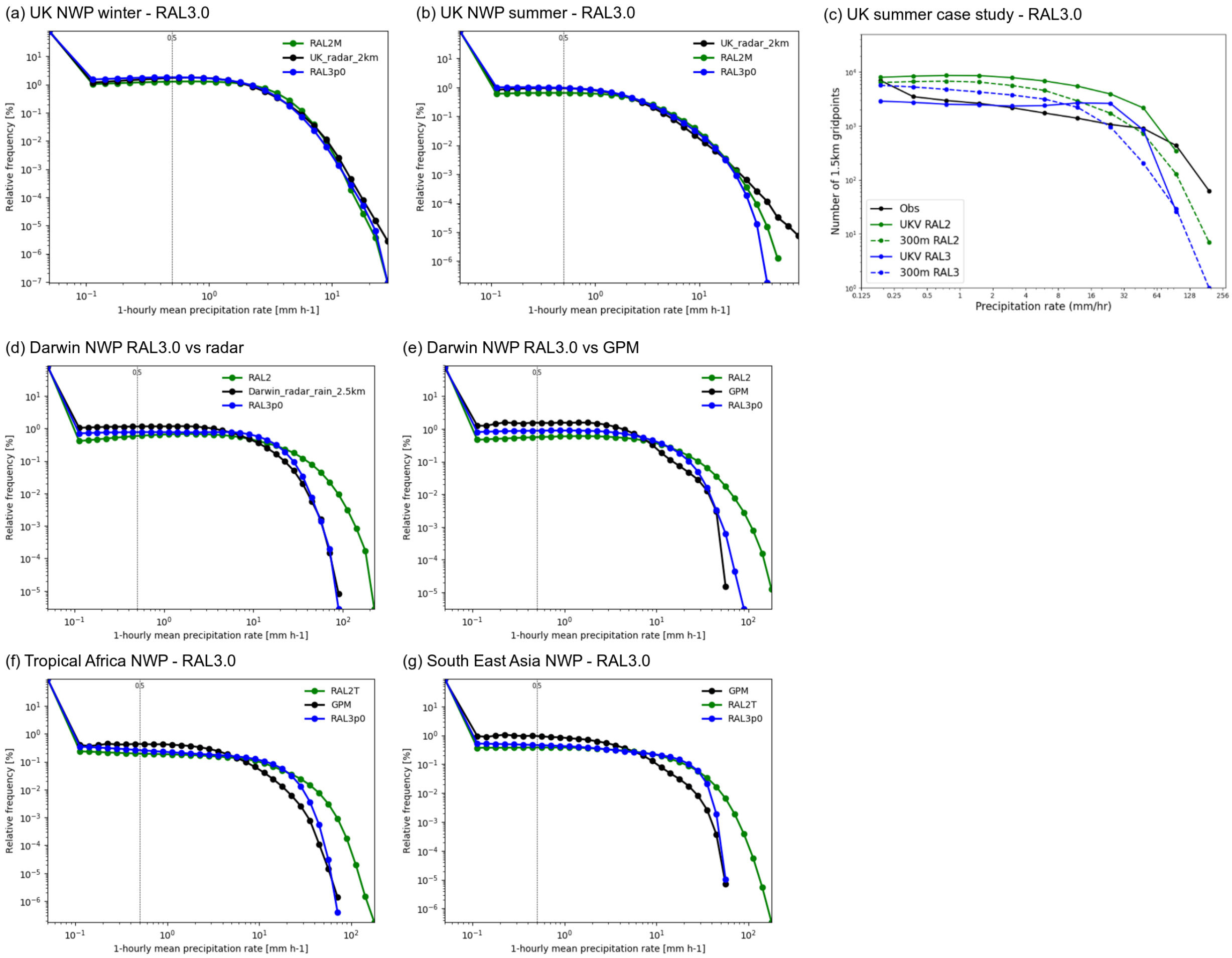

**Figure 4.** Hourly mean precipitation rate histograms for simulations (RAL3.0 results in blue and relevant RAL2 baseline in green) from NWP simulations for a) UK winter, b) UK summer, c) UK case study (19 July 2021) using grid lengths of 300 m and 1.5 km, d) Darwin, Australia relative to radar, e) Darwin relative to GPM IMERG V07B, f) Tropical Africa and g) South-East Asia. Black lines represent observations, either radar for UK domains and Darwin in (c) or GPM IMERG V07B for Darwin in (d) and other non-UK domains. In c), solid lines show 1.5 km grid UKV results, dashed lines show 300 m model results, with all data regridded to 1.5 km prior to analysis.

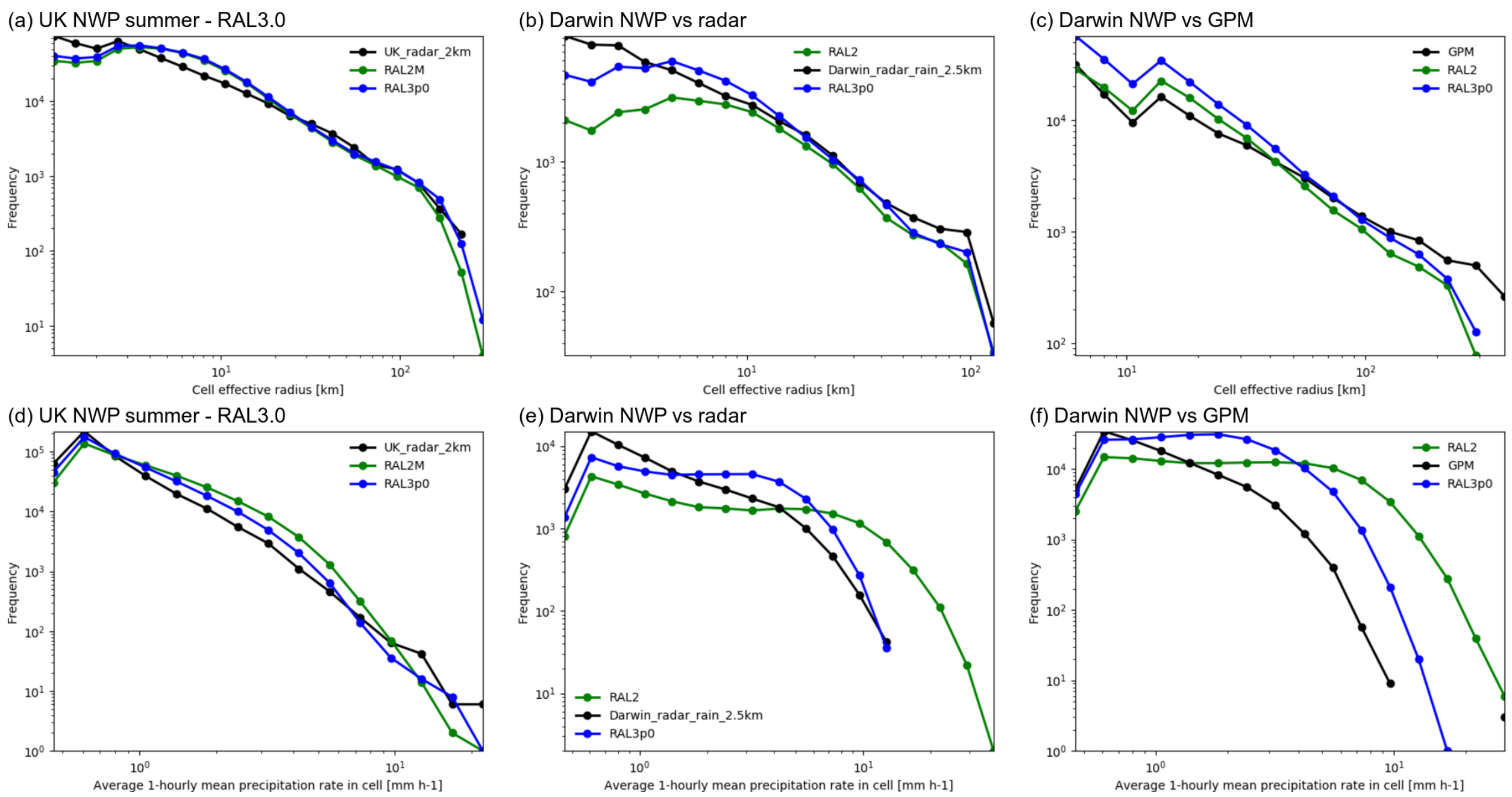

**Figure 5.** : Top row shows average size of precipitation cells for a) UK summer relative to radar, b) Darwin NWP relative to radar and c) Darwin NWP relative to GPM IMERG V07B. Bottom row shows average precipitation rate within storm cells for d) UK summer relative to radar observations, e) Darwin NWP relative to radar and f) Darwin NWP relative to GPM IMERG V07B. Observations are shown in black, green lines show relevant RAL2 baseline results and blue lines results for the initially released RAL3.0 configuration.

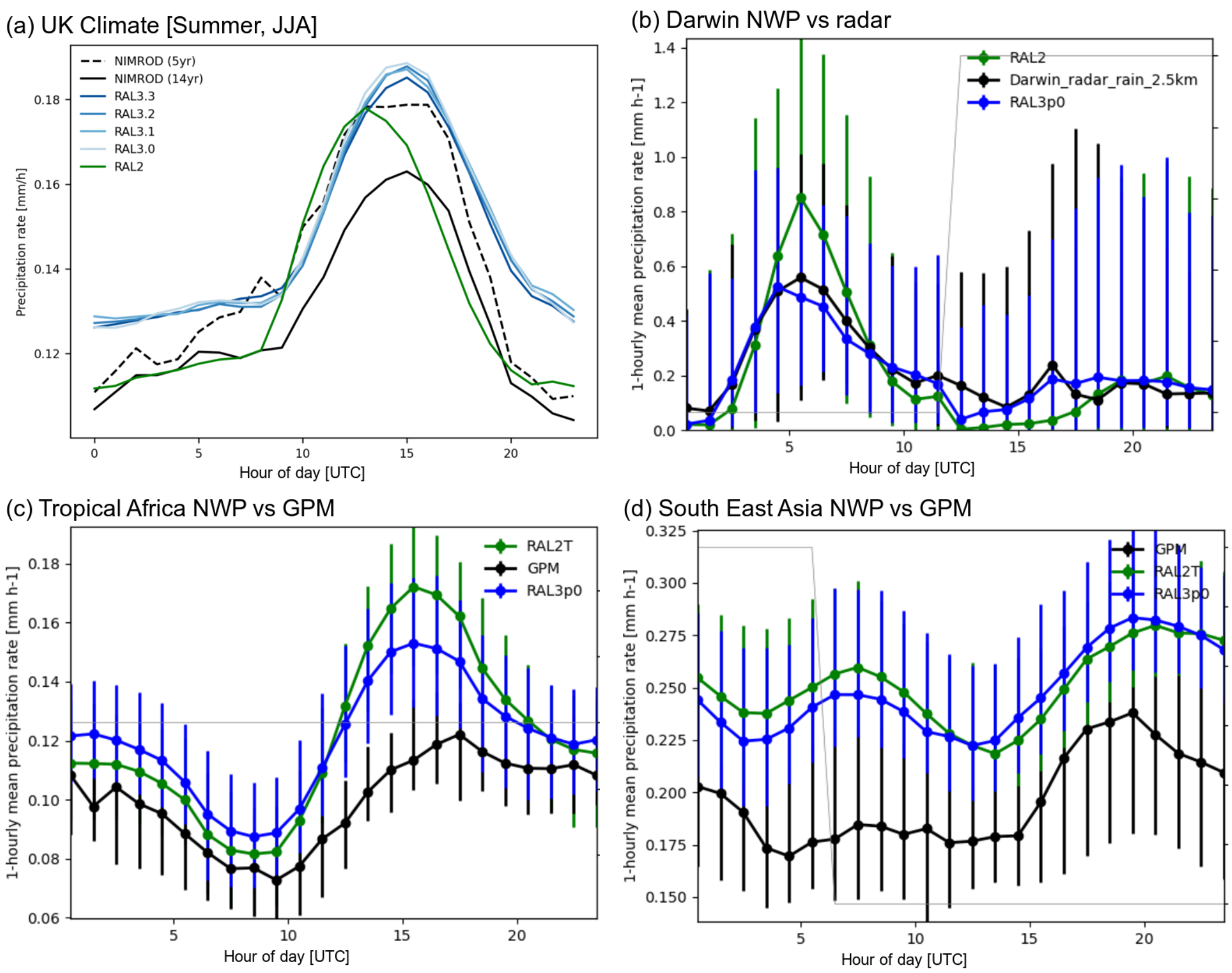

**Figure 6.** Diurnal cycles precipitation for a) 5-year UK Climate simulations in summer (JJA), averaged over the UK comparing RAL3 revisions with RAL2-M baseline, b) Darwin NWP relative to radar, c) Tropical Africa NWP relative to GPM IMERG V07B and d) domain-averaged South East Asia NWP relative to GPM IMERG V07B. Observations are shown in black, green lines show RAL2 results and blue lines results for RAL3 revisions. In a) observed diurnal cycles from radar observations (labelled NIMROD) are shown for the 14-year period 2003-2017 (solid black line), as well as for the same 5-year period as model runs. All time of day are shown in UTC.

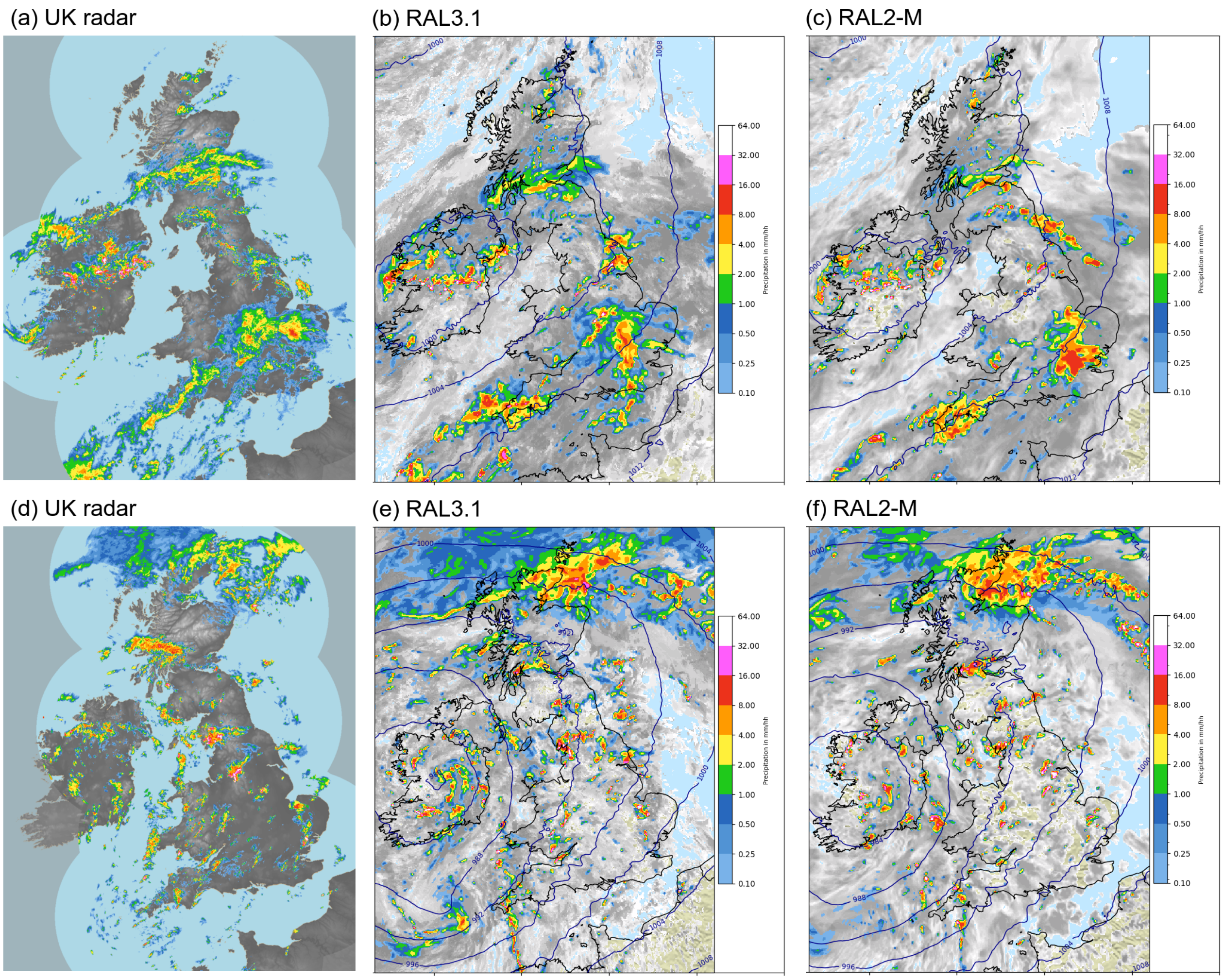

**Figure 7.** Illustrative snapshots of precipitation rate for two cases considered in subjective assessments of model performance over the UK comparing RAL3.1 and RAL2-M simulations with radar. (a-c) 1800 on 19 July 2019 for which RAL3.1 performed better than RAL2-M according to both Brier Score verification and unanimously preferred model output in subjective assessment. (d-f) 1800 on 9 August 2019 for which RAL2-M performed better according to Brier Score verification but the subjective assessment unanimously preferred RAL3.1 outputs for light precipitation. Model outputs are shown at 6h lead time (i.e. initialised at 1200).

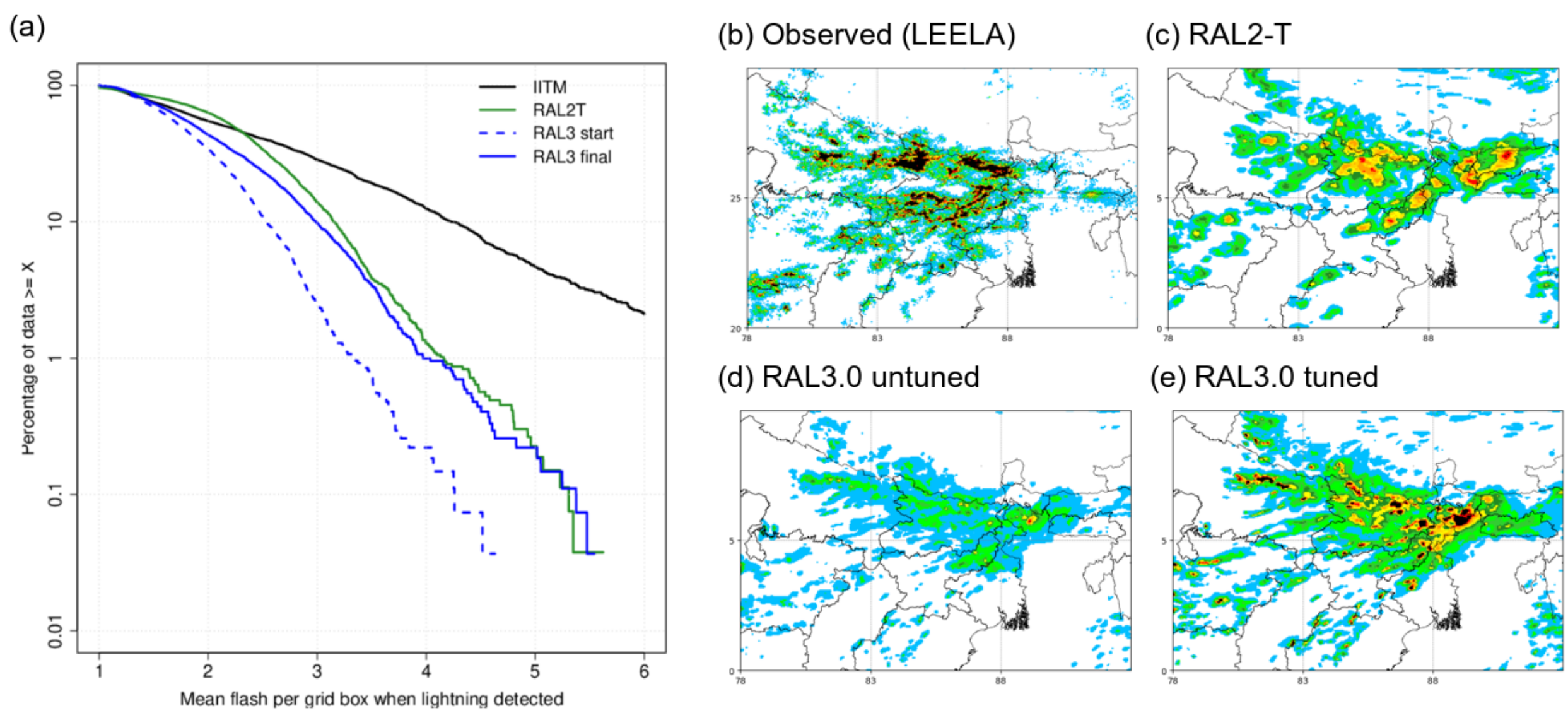

**Figure 8.** (a) Frequency distribution of mean flash rate per grid point with lightning detected as observed (black) and in variety of 4.4 km regional model simulations with different science configuration options. Maps show daily number of lightning flash counts for 25 June 2020 in northern India from (b) LEELA observations, (c) RAL2-T simulation, (d) RAL3.0 simulation using the same $k_1$ and $k_2$ parameters as used with RAL2-T (termed 'RAL3 start' in panel a)), (e) RAL3.0 simulation using updated parameter values (termed 'RAL3 final' in panel a).

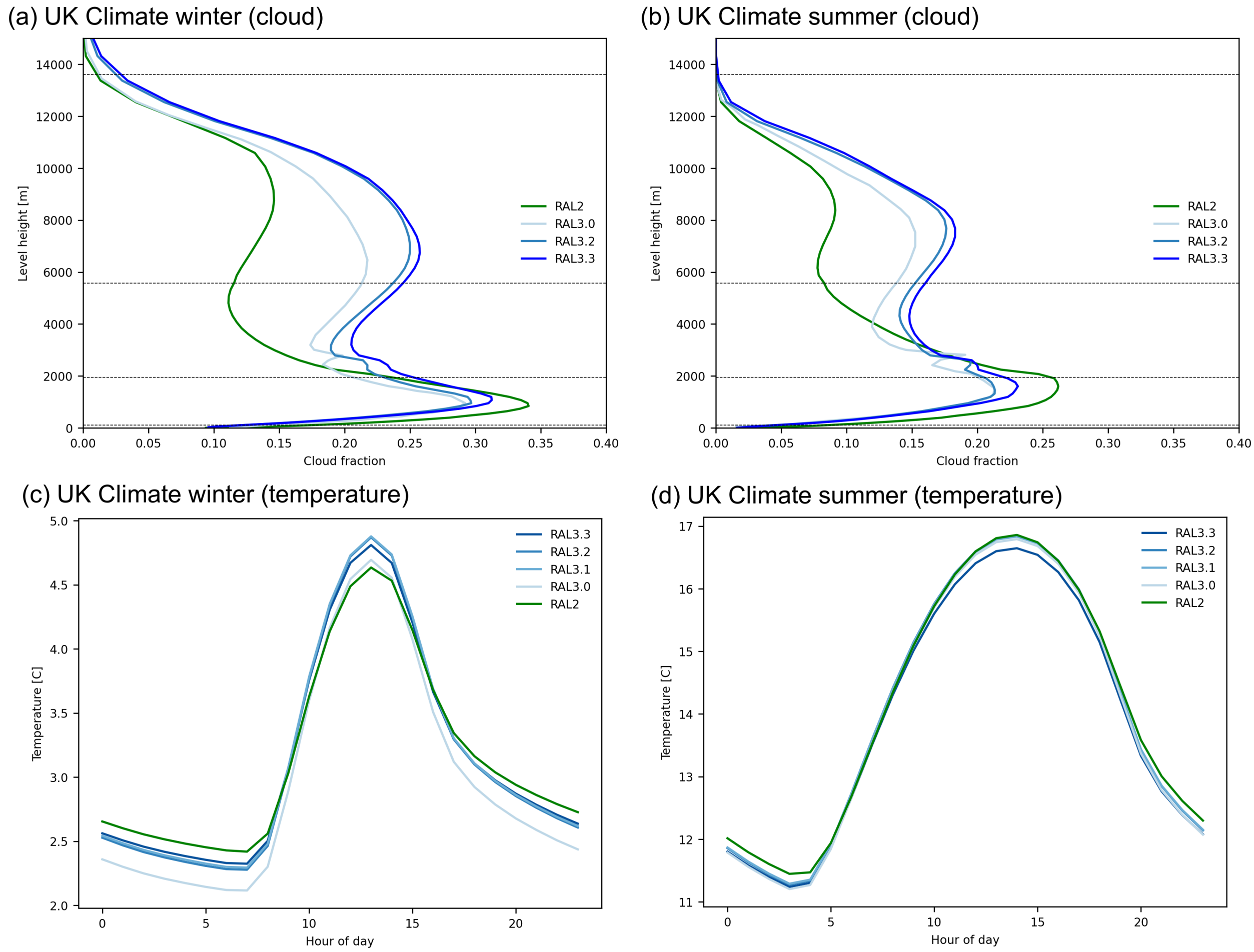

**Figure 9.** Mean cloud and temperature results from 5-year UK climate simulations. (a) Mean cloud fraction profile with height for winter (DJF), and (b) mean cloud profiles for summer (JJA). (c) Mean diurnal cycle of near-surface (1.5 m) temperature for winter (DJF) and (d) mean diurnal cycle for summer (JJA). Green lines represent RAL2-M, and blue lines show RAL3 across successive revisions. Horizontal lines in panels (a) and (b) represent upper boundaries for low, medium, high cloud and very high cloud diagnostic definitions. Note that the spike in cloud cover at 3000 m in RAL3.0 is due to a bug in the bimodal cloud scheme code that has since been rectified from RAL3.1 revision onwards.

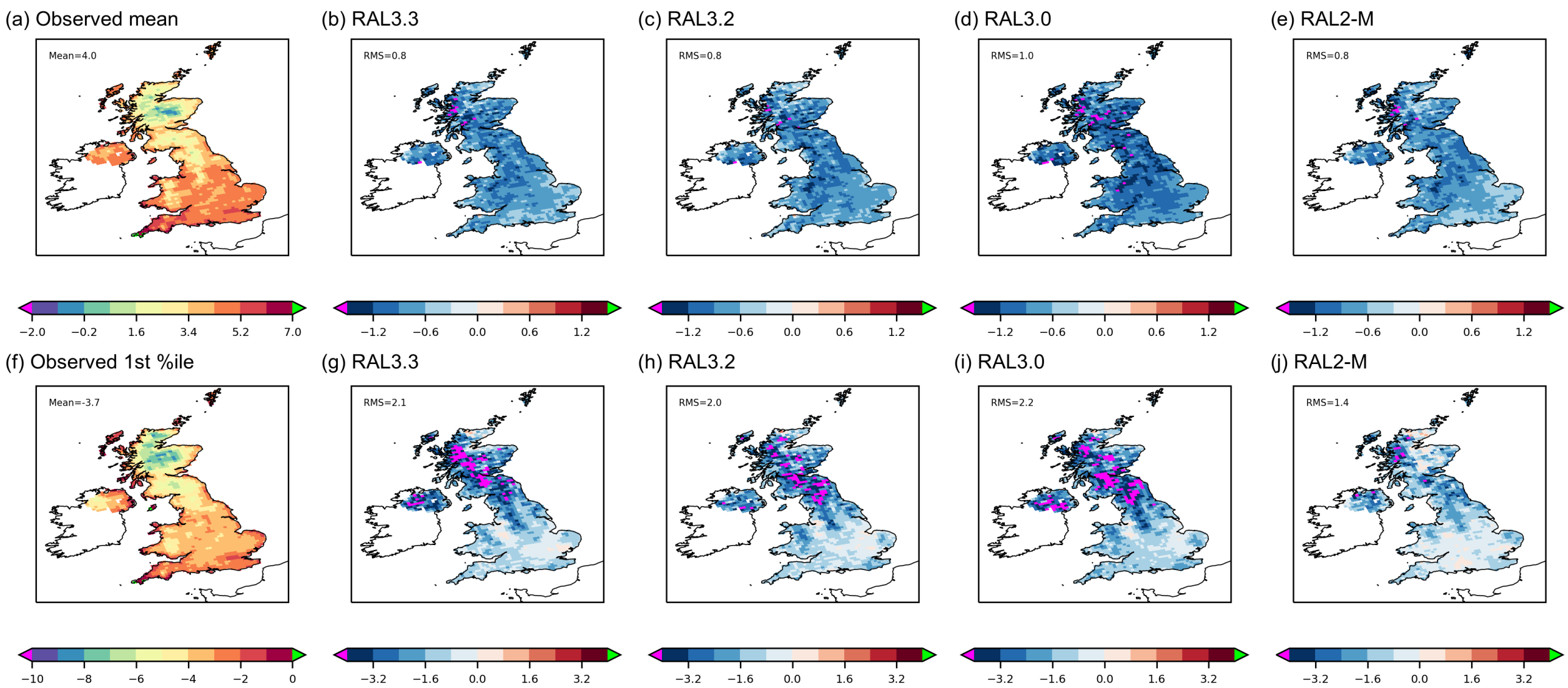

**Figure 10.** UK climate near-surface (1.5 m) temperature mean and 1st percentile during winter, DJF period from 5-year simulations. (a) Observed mean from gridded observations, and anomalies relative to observations of mean temperature simulated using (b) RAL3.3, (c) RAL3.2, (d) RAL3.0 and (e) RAL2-M. (f) Observed 1st percentile (coldest temperatures), and anomalies relative to observations of 1st percentile simulated using (g) RAL3.3, (h) RAL3.2, (i) RAL3.0 and (j) RAL2-M.

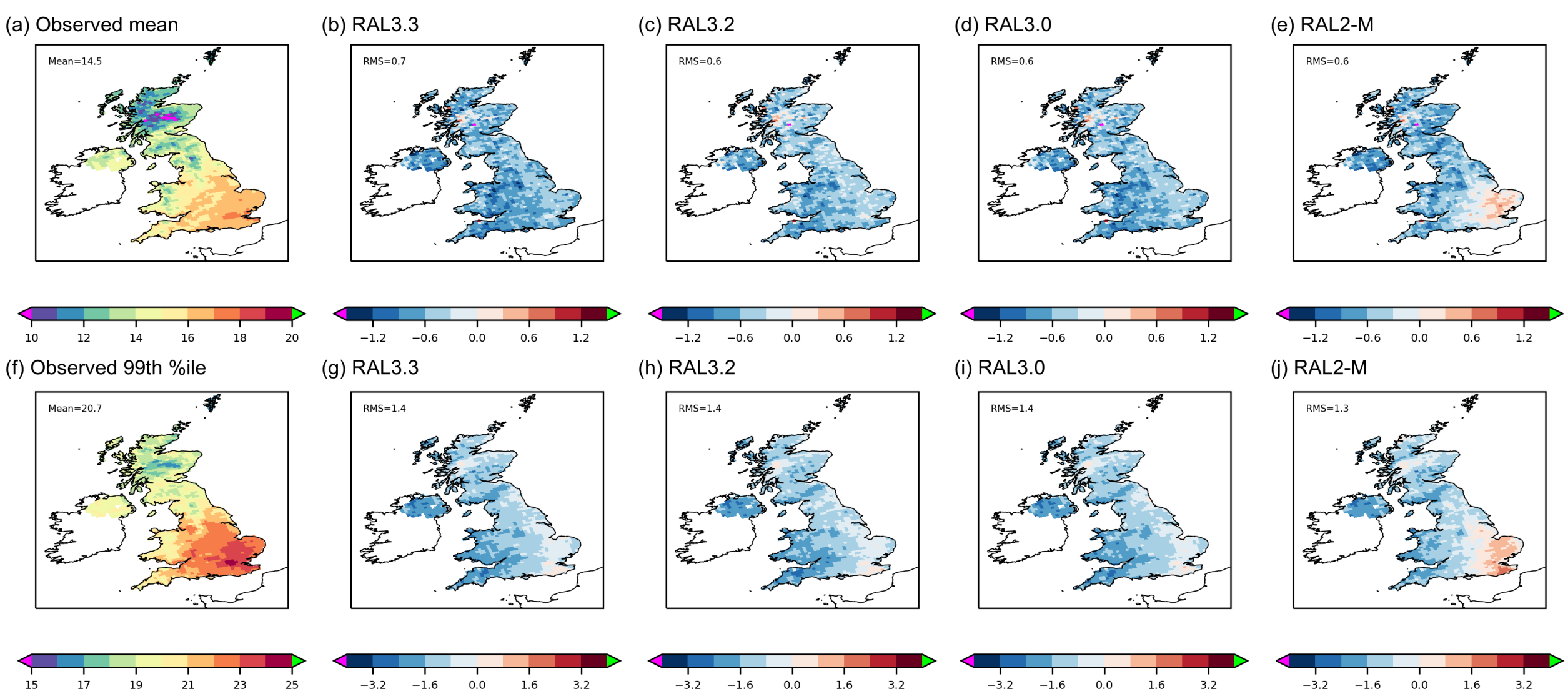

**Figure 11.** UK climate near-surface (1.5 m) temperature mean and 1st percentile during summer, JJA period from 5-year simulations. (a) Observed mean from gridded observations, and anomalies relative to observations of mean temperature simulated using (b) RAL3.3, (c) RAL3.2, (d) RAL3.0 and (e) RAL2-M. (f) Observed 99th percentile (warmest temperatures), and anomalies relative to observations of 99th percentile simulated using (g) RAL3.3, (h) RAL3.2, (i) RAL3.0 and (j) RAL2-M.

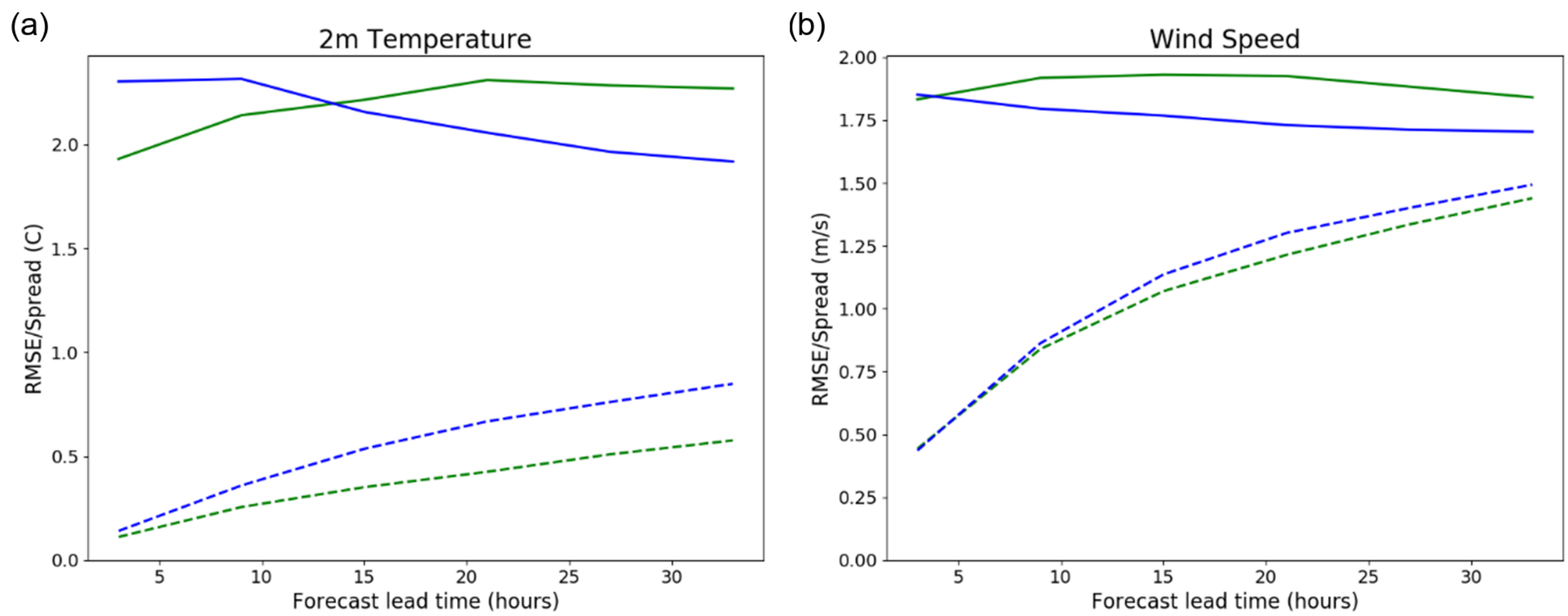

**Figure 12.** Ensemble spread (dashed line) and error (solid line) for the RAL2-T (green) and RAL3.0 (blue) ensembles run over the Darwin domain. Results are for a 10-day ensemble trial during local summer from 29 November to 8 December 2020.

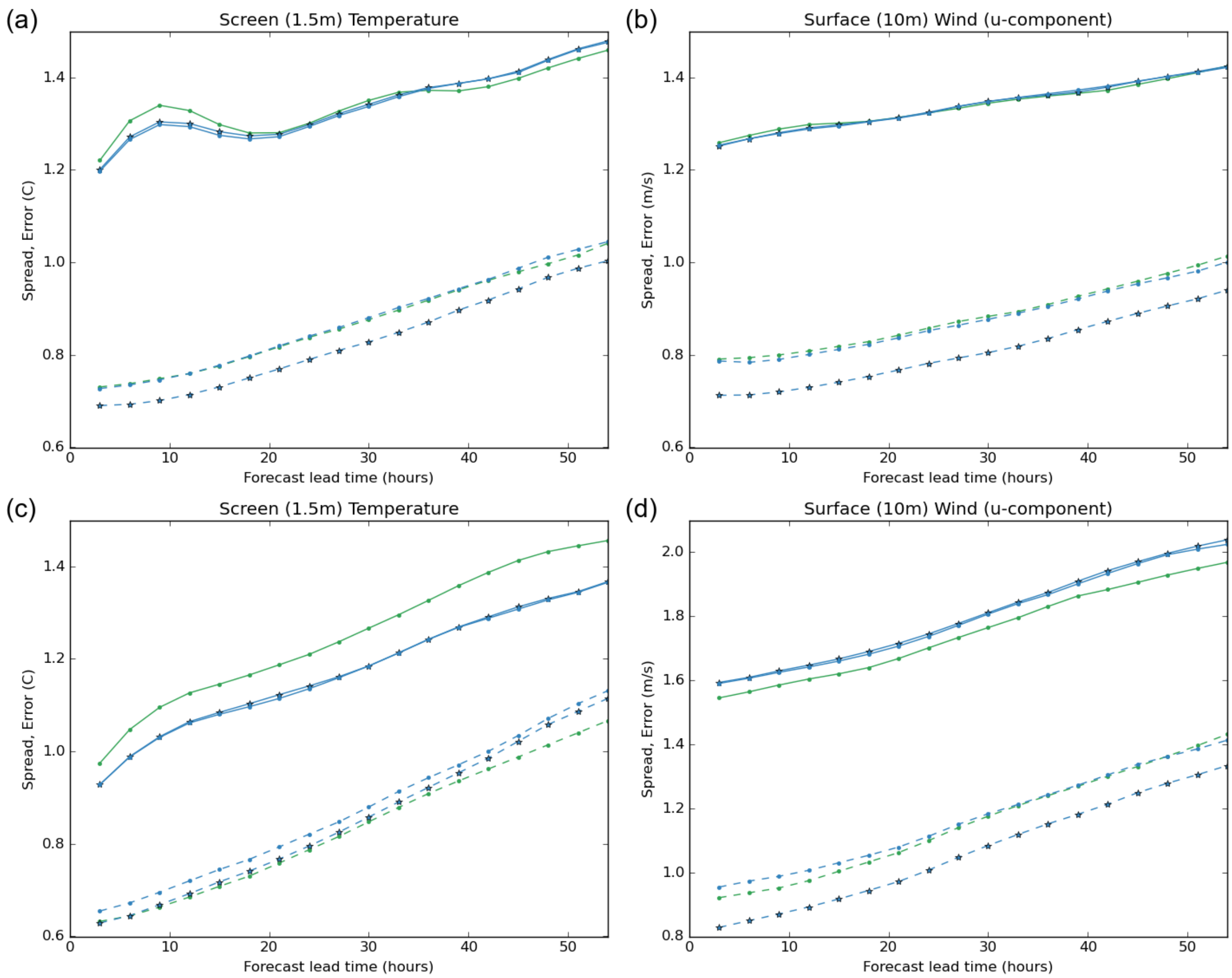

**Figure 13.** Ensemble spread (dashed line) and error (solid line) of the RAL2-M (green), RAL3.3 (blue, starred) and RAL3 with new random parameters (blue) ensembles over the UK domain. The top row shows the summer trial period (20220715 to 20220815) and the bottom row shows the winter trial period (20211201 to 20220101). Note the RAL2-M and RAL3 ensembles have different driving models (current global operational GC4 ensemble driving RAL2-M and next-global GC5 configuration ensemble driving RAL3.3); the only difference between the two RAL3 ensembles are the new random parameters.

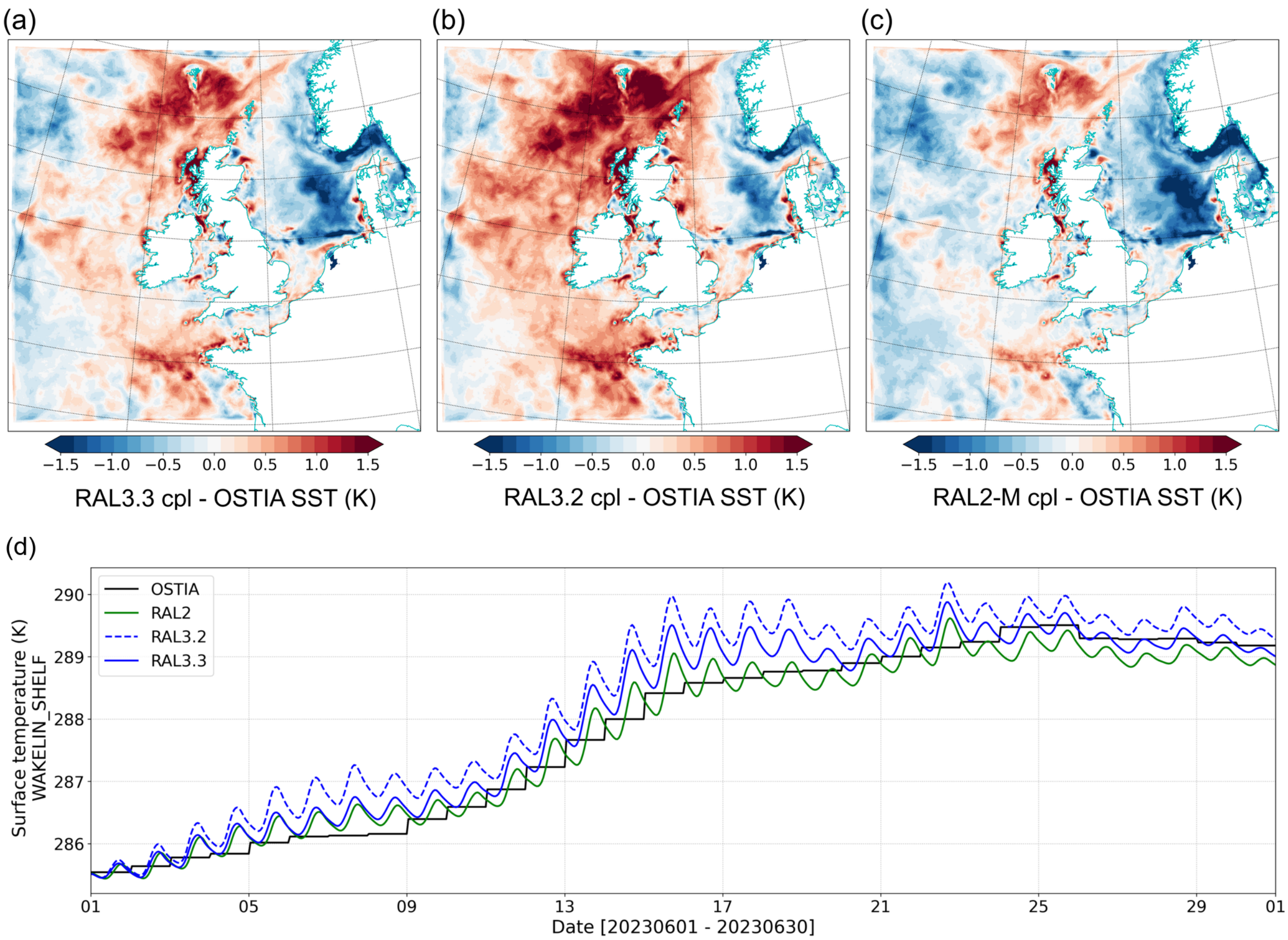

**Figure 14.** Difference in mean surface temperature (top ocean model level temperature) at 0600 UTC during June 2023 simulated by a regional ocean model coupled to a CP atmosphere using (a) RAL3.3, (b) RAL3.2 and (c) RAL2-M. Differences are shown relative to the mean daily foundation SST derived from observations in OSTIA. (d) Timeseries of simulated SST and daily OSTIA during June 2023 as average across northwest shelf region.

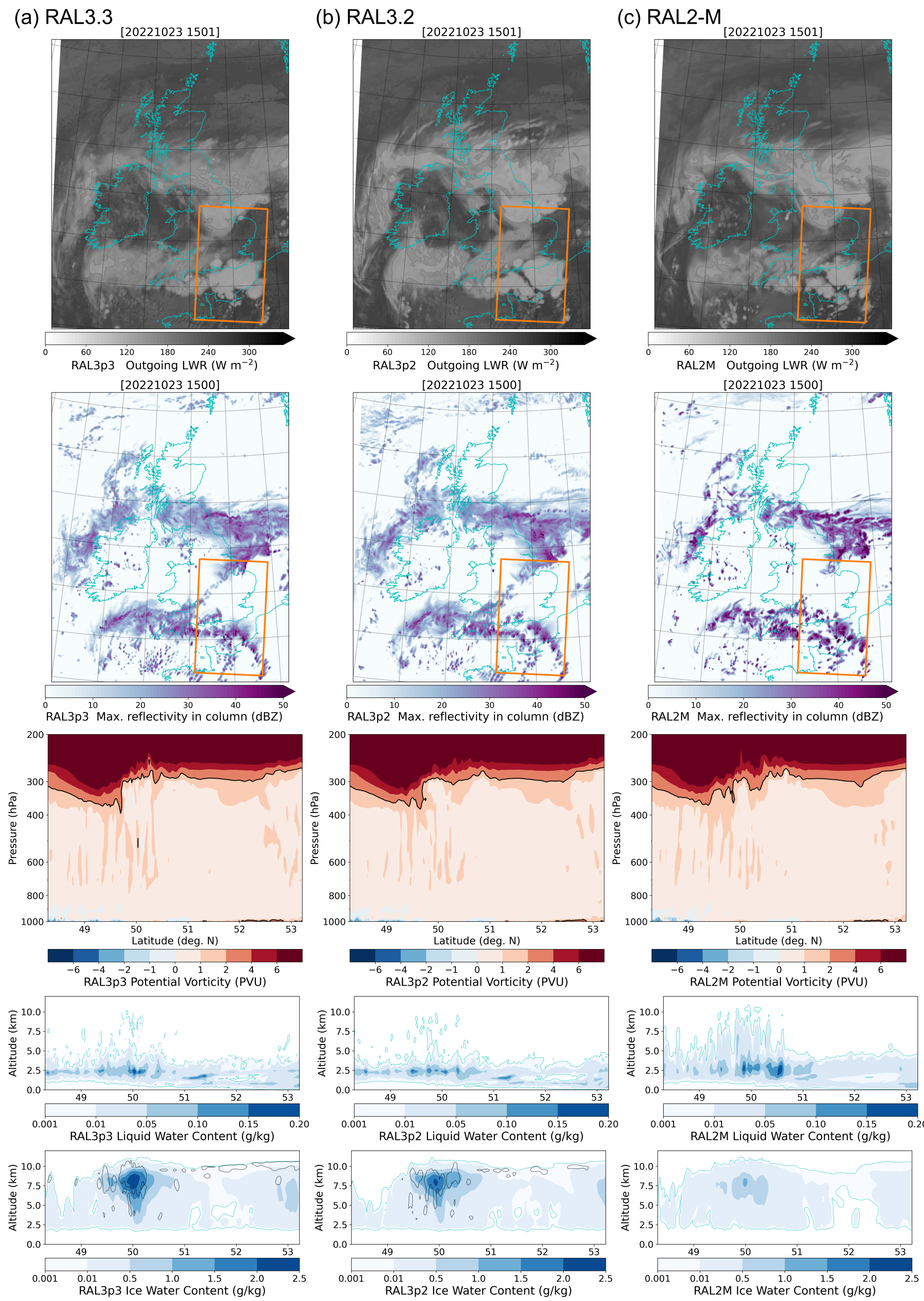

**Figure 15.** Illustration of RAL3 and RAL2-M results for a severe quasi-linear convective system over south-East England on 23 October 2023. All results are shown for 1500 UTC, and all simulations use the same lateral boundary conditions, and run at UM version 13.5. Map plots shows instantaneous outgoing longwave radiation and maximum reflectivity in the vertical column for a sub-region of the model domain. The lower 3 rows show average latitude-height cross-sections through the orange box marked in upper panels. These show simulated potential vorticity (as function of pressure up to 200 hPa), liquid water content and ice water content (both as function of altitude in the lowest 12 km only). Black contours in the lower panel indicate the presence of cloud ice crystals as a diagnostic from CASIM.