# Peer review of "The third Met Office Unified Model-JULES Regional Atmosphere and Land Configuration, RAL3"

_Geoscientific Model Development, 2024_

## Author Response (AR1)

GMD-2024-201: Author's response including a point-by-point response (*in red italics*) to the reviews including a list of all relevant changes made in the manuscript. Response date: 10/02/2025.

**Reviewer 1**

• Certain details, such as repository ticket numbers, might be overly specific and could be reconsidered for brevity.

**Thank you very much for your comments.**

Regarding the repository ticket numbers, this is a key part of the documentation and quality assurance of the Regional Atmosphere and Land development process. Therefore, although it does add to the length of the manuscript, we think it is necessary to keep this information in the paper. In addition, we feel that it can potentially provide very useful information for the reader.

**Reviewer 2**

Specific comments

• P3L25: The sentence "While CP climate models ... with parameterized convection (Kendon et al., 2017)" is rather long. I suggest breaking it into two sentences for improved readability: "While CP climate models do not necessarily better represent daily mean precipitation (e.g., Berthou et al., 2020), they typically show improved sub-daily rainfall characteristics. These include better representation of the diurnal cycle of convection, the spatial structure of rainfall, duration-intensity characteristics, and the intensity of hourly precipitation extremes compared to climate models with parameterized convection (Kendon et al., 2017)."

Made changes as suggested by the reviewer

• P4L6: There is a repeated word: "inform inform"

**Made changes as suggested by the reviewer**

• P6, section 2.3: What is the minimum model grid spacing, the full \$dz\$? Is it 5 or 10 m? What is the maximum grid spacing at the model top?

Added the following text: "For scalar variables this means that the minimum layer thickness is 5m (surface to level 1) and the maximum layer thickness is 1327m (level 89 to level 90)"

• Table 1: At this point in the manuscript, it is not clear what "Murk aerosol" is. I suggest to add a reference.

**Added reference to Clark et al (2008)**

Clark, P. A., Harcourt, S. A., Macpherson, B., Mathison, C. T., Cusack, S., and Naylor, M.: Prediction of visibility and aerosol within the operational Met Office Unified Model. I: Model formulation and variational assimilation, Quarterly Journal of the Royal Meteorological Society, 134, 1801–1816, https://doi.org/10.1002/qj.318, http://doi.wiley.com/10.1002/qj.318, 2008.

• P9L11: "... and it uses a fixed in-cloud number concentration". Of what? Aerosols? CCNs?

Added the clarifying word "droplet"

"and it uses a fixed in-cloud droplet number concentration"

• P9L28: Change "so less well" to "so it is less well".

Made changes as suggested by the reviewer

• P12L1-4: The last sentence is not clear to me. What exactly is your statement? How can a changed diagnostic influence the behavior of the turbulence scheme?

We thank the reviewer for this comment and agree that this was not well explained as there is no impact on the turbulence scheme. We have rewritten this section as follows to make it clearer:

"Turbulent kinetic energy (TKE) and variance diagnostics have been revised (RAL Ticket #87). Diagnostics for TKE and variances of vertical velocity, temperature and humidity include terms proportional to the scalar fluxes. In RAL3 we now use those fluxes directly rather representing them as a down-gradient diffusion. As well as being more accurate this avoids an issue where the non-local BL scheme parametrizes entrainment fluxes across sharp inversions explicitly and sets the diffusion coefficients there to zero, thus returning zero TKE in RAL2M".

• P17L7: "targetted" à "targeted"

Made changes as suggested by the reviewer

• P23L18: Check the reference "P et al., 2021", also in the references on P38.

Made corrections as suggested by the reviewer. Reference is:

Barrett, P., Abel, S., Lean, H., Price, J., Stein, T., Stirling, A., and Darlington, T.: WesCon 2023: Wessex UK Summertime Convection Field Campaign, https://doi.org/10.5194/egusphere-egu21-2357, https://doi.org/10.5194/egusphereegu21-2357, 2021.

• P28L18-L25: What is SINGV?

Added two references: Heng et al., 2020; Dipankar et al., 2020.

Heng, B. C. P., Tubbs, R., Huang, X.-Y., Macpherson, B., Barker, D. M., Boyd, D. F. A., Kelly, G., North, R., Stewart, L., Webster, S., and Wlasak, M.: SINGV-DA: A data assimilation system for convective-scale numerical weather prediction over Singapore, Quarterly Journal of the Royal Meteorological Society, 146, 1923–1938, https://doi.org/https://doi.org/10.1002/qj.3774, 2020.

Dipankar, A., Webster, S., Sun, X., Sanchez, C., North, R., Furtado, K., Wilkinson, J., Lock, A., Vosper, S., Huang, X.-Y., and Barker, D.: SINGV: A convective-scale weather forecast model for Singapore, Quarterly Journal of the Royal Meteorological Society, 146, 4131–4146, https://doi.org/https://doi.org/10.1002/qj.3895, 2020

• Table 3: What is CLASSIC? To clarify, but "EasyAerosol" on a new line.

Made changes as suggested by the reviewer

• Figure 6: What do the bars represent? Standard deviation? Minimum to Maximum?

Added the following text to the Figure 6 caption: The error bars on plots b) to d) enclose the interquartile range (25th and 75th percentile values)

**Reviewer 3**

**Minor Suggestions:**

• On line 6, page 4, change "inform inform" to "inform."

Made changes as suggested by the reviewer

• The authors claim that cloud bases and fractions have been improved across RAL3.0 and 3.3. It would be useful to add a cloud verification in Figure 9 to support this.

We are not aware of any observations of vertical profiles of cloud fraction that we could easily add to Figure 9. Therefore, we just refer the reviewer to Figure 2 panels (a) and (b) regarding HiRA plots for cloud base/fraction verification over the UK.

There may be some satellite data (IR sounders) that could be used, but then the cloud fraction is not necessarily in same format as how model calculates cloud fraction. We acknowledge that this is possibly something to consider for future RAL cycles.

• It would be helpful to include composite radar reflectivity observations in either Figure 15 or S4.

The focus of this section is on understanding the differences in processes between the models, not on whether the forecast was correct. Therefore, for brevity a radar composite was not included. We do not feel that the inclusion would impact the reader's understanding of this section and so have decided not to include it to save space in what is already a long paper.